# Neuronal representation of saccadic error in macaque posterior parietal cortex (PPC)

**Yang Zhou[1,2,3], Yining Liu[4], Haidong Lu[1], Si Wu[1], Mingsha Zhang[1]\***

[1]State Key Laboratory of Cognitive Neuroscience and Learning, Beijing Normal University, Beijing, China; [2]Institute of Neuroscience, Shanghai Institutes for Biological Sciences, Chinese Academy of Sciences, Shanghai, China; [3]University of Chinese Academy of Sciences, Shanghai, China; [4]The First Affiliated Hospital of Zhengzhou University, Zhengzhou, China

**Abstract** Motor control, motor learning, self-recognition, and spatial perception all critically depend on the comparison of motor intention to the actually executed movement. Despite our knowledge that the brainstem-cerebellum plays an important role in motor error detection and motor learning, the involvement of neocortex remains largely unclear. Here, we report the neuronal computation and representation of saccadic error in macaque posterior parietal cortex (PPC). Neurons with persistent pre- and post-saccadic response (PPS) represent the intended end-position of saccade; neurons with late post-saccadic response (LPS) represent the actual end-position of saccade. Remarkably, after the arrival of the LPS signal, the PPS neurons' activity becomes highly correlated with the discrepancy between intended and actual end-position, and with the probability of making secondary (corrective) saccades. Thus, this neuronal computation might underlie the formation of saccadic error signals in PPC for speeding up saccadic learning and leading the occurrence of secondary saccade.

**\*For correspondence:** mingsha. zhang@bnu.edu.cn

**Competing interests:** The authors declare that no competing interests exist.

## Introduction

Interacting accurately with the environment is critical for an animal's survival. However, a noticeable fact is that the executed actions are not always perfectly matched with the intended (desired) ones, but have errors. This is true even for very well trained actions (*Shadmehr et al., 2010*). Therefore, the brain must evolve efficient mechanisms to detect the motor errors between the intended and the executed actions for motor control, motor learning, spatial perception, and other motor-related cognitive functions.

The executed motor actions can be encoded by two types of signals in the brain: the internal movement-related signals (e.g., efference copy/corollary discharge) (*Helmholtz, 1925*; *Sherrington, 1918*) and the external sensory signals (e.g., proprioception) (*Wang et al., 2007*). In self-originated movements, the internal signals predict the displacement of motor effector (*Crapse and Sommer, 2008*; *Duhamel et al., 1992*; *Helmholtz, 1925*; *Sommer and Wurtz, 2002*; *Sperry, 1950*), whereas the external sensory signals inform its actual position (*Fuchs and Kornhuber, 1969*; *Sherrington, 1918*). Because the emergence of the sensory information lags the execution of the movement, it is believed that the motor errors in high velocity movements, such as saccades, are calculated by relying heavily on the internal signals (*Kawato, 1999*; *Robinson, 1975*; *Shadmehr and Krakauer, 2008*; *Shadmehr et al., 2010*; *Wolpert et al., 1995*). Over the past three decades, the online control of saccades as well as saccadic learning can, in principle, be fully explained by the well-established models that reflect the properties of the brainstem-cerebellum machinery for

saccades (*Barash et al., 1999*; *Catz et al., 2005*; *Dash and Thier, 2014*; *Ito, 1970*; *Kawato et al., 2003*; *Nowak et al., 2007*; *Pasalar et al., 2006*; *Robinson, 1975*; *Shadmehr and Krakauer, 2008*; *Shadmehr et al., 2010*; *Stein, 2009*; *Wolpert et al., 1998*). However, up to date, whether neocortex is also involved in saccadic error detection is largely unknown.

We recently recorded the activity of a single neuron from posterior parietal cortex (PPC) of two monkeys while they were performing oculomotor tasks. Unexpectedly, we found the neuronal computation and representation of the saccadic error. One group of PPC neurons discharged persistently before and after saccades (PPS neurons). Another group of PPC neurons discharged purely post-saccadically (LPS neurons), starting to discharge ~70 ms after the completion of the saccade. A correlation analysis between the neuronal discharge and the end-position of the saccades suggested that the PPS neurons represented the intended end-position of the saccade (i.e., the location of the visual target) whereas the LPS neurons represented the actual end-position of the saccade. The PPS neurons started to decay their activities shortly after the increase of LPS neurons' activities, fitting the temporal request for comparing the motor intention and sensory input (*Figure 1A*). Furthermore, the activity of the PPS neurons shortly after the arrival of the LPS signal was highly correlated with the magnitude of saccadic error and the probability of making a secondary saccade. Interestingly, the activity level of PPS neurons during this period resembled the subtraction between the intended and the actual end-position signals.

Taken together, our results showed that the PPS neurons in PPC behaved like an error detector that computed the saccadic error by comparing the intended and the actual saccade end-position signals. This error signal might be used for speeding up saccadic learning and predicting the occurrence of secondary saccade.

## Results

The main behavioral task in the present study is named as spatial-cue delayed saccade task (SCS, *Figure 1B*). In this task, monkeys chose one from two identical visual stimuli (left versus right visual field) as the saccadic target, based on the position of an additional visual cue either appearing to the left or right of the visual field. The memory-guided saccade task (MGS, *Figure 1C*) was used to find the lateral intraparietal area (LIP) based on its physiological signature: neurons discharged persistently throughout the memory interval (*Andersen and Buneo, 2002*; *Snyder et al., 1997*; *Zhang and Barash, 2004*). In total, we recorded 753 neurons, in which 377 out of 753 neurons showed significant pre-saccadic activity (pre-saccadic neurons) whereas 107 neurons showed significant pure post-saccadic activity (post-saccadic neurons). The neuronal data shown here were recorded from the same holes (0.5 mm in diameter for each hole and 0.5 mm between two holes) of a recording grid in which we recorded persistent response neurons in the MGS task. The reconstructed recording sites show that all neurons, but one, were recorded within a small area of 3x4 mm$^2$ in monkey B and 3x3 mm$^2$ in monkey D. Therefore, the neurons shown in this study were probably recorded mostly from the LIP. We identified a group of pre-saccadic response neurons (89 out of 377) that discharged persistently in both pre- and post-saccadic intervals (PPS) when saccades were directed to the response fields (preferred direction). We also identified another 27 neurons with late post-saccadic responses (LPS).

### The perisaccadic activity of PPS neurons reflects the intended end-position of saccade

The activity of an example PPS neuron in the spatial-cue delayed saccade (SCS) is shown in *Figure 2A*. In the preferred direction (black), the neuron started to increase in activity ~200 ms prior to the initiation of the saccade, and reached peak activity at ~100 ms after the end of the saccade. Then it started to decay in activity. Similar firing pattern was seen in the population activity of 89 PPS neurons (*Figure 2B*). On average, the activity of PPS neurons started to decay at ~74.8 ms (STD = 33.1) after the completion of the saccade.

The pre-saccadic activity of the PPS neurons was strongly correlated with the direction of saccades. The relationship between the pre-saccadic activity (-200~0 ms before saccade start) and the direction of saccades (left versus right) is shown in *Figure 2C*. All but seven neurons had significantly higher activity (p<0.05, Wilcoxon test) in the preferred direction than in the non-preferred one. We further analyzed the correlation between the early post-saccadic activity (0–100 ms after saccade

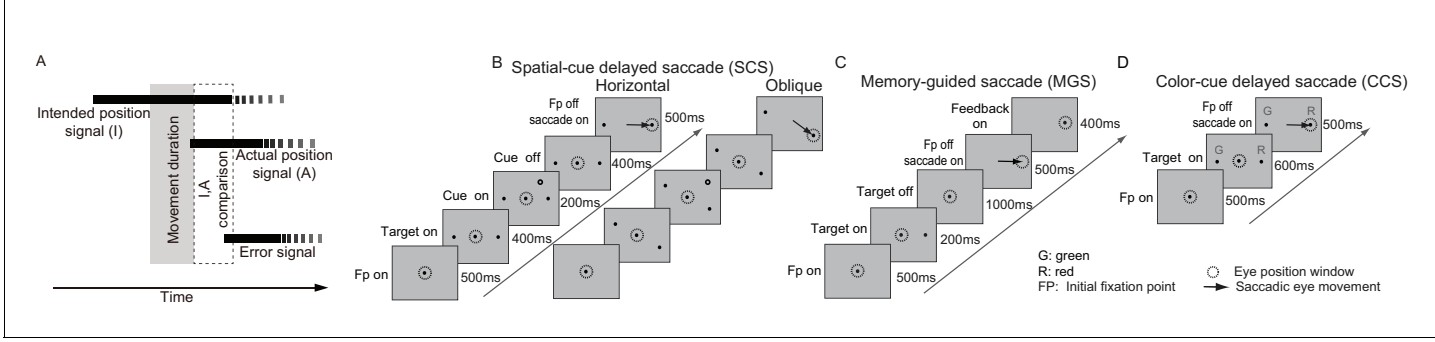

**Figure 1.** Conceptual model and behavioral paradigms. (**A**) The proposed temporal sequence for comparison of the intended and the actual position signals. The intended signal raises up before the initiation of movement and lasts after the arrival of the external sensory signals. The external sensory signal raises up after the start of movement. The comparison occurs after the convergence of these two signals (marked by the dashed rectangle). The error signal is generated after the comparison. (**B**) Spatial-cue delayed saccade task (SCS). Horizontal direction trials: target is 10° in eccentricity. Oblique direction trials: target is 13° in eccentricity. (**C**) Memory-guided saccade task (MGS). (**D**) Color-cue delayed saccade task (CCS).

end) and the pre-saccadic activity of PPS neurons. The distribution of correlation coefficient values based on single trial analysis of 89 PPS neurons is shown in *Figure 2—figure supplement 1*. The results show that the pre-saccadic and the early post-saccadic activities were significantly and positively correlated in 34 of 89 neurons (p<0.05, t-test), and this correlation was also observed in the population data (mean r = 0.2151, p<0.001, paired t-test). Such results imply that the early post-saccadic activity of PPS neurons might encode similar information as the pre-saccadic activity did. Therefore, we introduced an epoch, the perisaccadic interval (-150~100 ms of saccade end), pooling the pre- and the early post-saccadic intervals.

We then assessed whether the perisaccadic activity of PPS neurons consistently encoded the actual post-saccadic eye position. To do so, we analyzed the correlation between the perisaccadic activity and the end-position of saccade for each PPS neuron on single trial basis. 55 out of 62 PPS neurons were recorded with enough trials (>30 trials in the preferred direction) that allowed us to make the correlation analysis when the monkeys performed the horizontal SCS task (*Figure 1B*). Overall, the correlation coefficient values of these 55 PPS neurons were symmetrically distributed around zero (*Figure 2D*, mean r = 0.0132, p=0.5242, paired t-test), suggesting that their responses are not directly encoding the actual end-position of saccades. On a single neuron level, most PPS neurons (45 out of 55) were uncorrelated with the end-position of saccades, with the remaining 10 neurons reaching statistical significance (p<0.05) but having inconsistent mixed positive and negative correlation: 4 were negatively correlated and 6 positively. Similar results were observed in 24 out of 27 PPS neurons that had enough trials for correlation analysis in the oblique SCS task (*Figure 1B*). There was no significant correlation between the perisaccadic activity and the end-position of saccade either in the horizontal (*Figure 2—figure supplement 2A*, mean r = −0.0209, p=0.4240, paired t-test) or in the vertical (*Figure 2—figure supplement 2A*, mean r=0.0170, p=0.5245, paired t-test) direction of saccade end-positions. These results indicated that the perisaccadic activity of PPS neurons does not likely encode the actual end-position of saccades. However, results are consistent if PPS does code for the intended end-position of the saccade.

There might be other factors that could possibly affect the early post-saccadic activity of PPS neurons. For instance, since the saccadic targets remained on until the end of a trial in the SCS task (*Figure 1B*), it is possible that the early post-saccadic activity of PPS neurons reflected the foveal visual stimulation, which followed the change in gaze direction from the central fixation point to the saccadic target. However, we found that during the MGS task, the same PPS neurons were not visually responsive during the visual feedback after finishing saccades (*Figure 2—figure supplement 3*). Note that in the MGS task, the target reappeared ('feedback on' in *Figure 1C*) after the saccade (273 ms in average). The absence of a visual response during the MGS feedback stimulus suggests that the perisaccadic activity of PPS neurons was not a foveal visual response.

Moreover, the observed variation in the perisaccadic activity for preferred and non-preferred directions cannot be caused by different reward signals. This is because the reward was identical

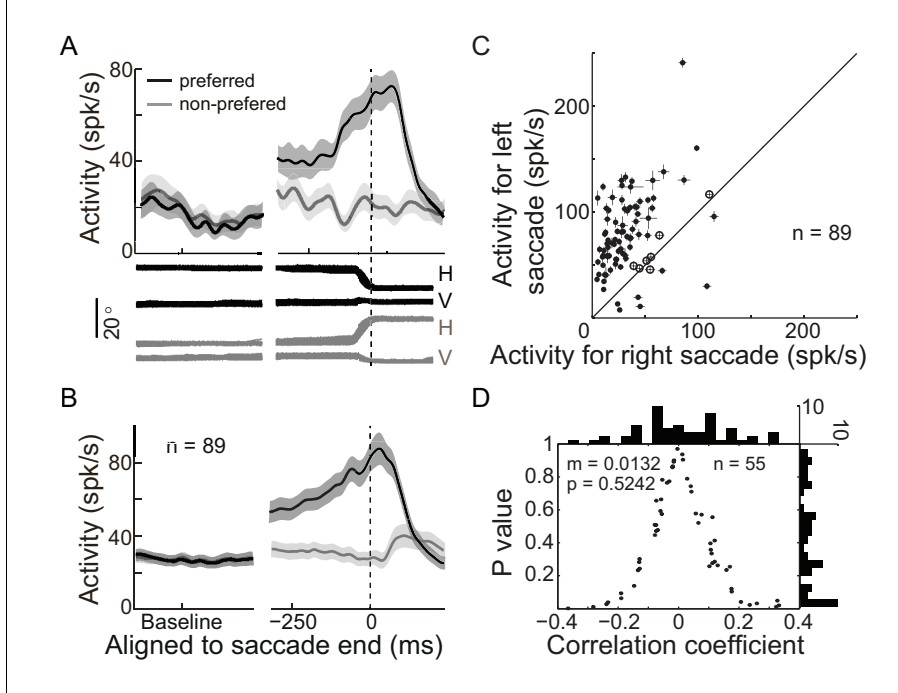

**Figure 2.** PPS neurons represent the intended end-position of saccades. (**A**) The activity of an example PPS neuron in the SCS task. Averaged spike density with the 95% confidence interval (shaded area) in the preferred (black) and null direction (grey) are shown in the upper panel, whereas the horizontal and vertical eye traces are shown in the lower panel. (**B**) The averaged population activity of 89 PPS neurons (69 neurons from monkey B, 20 neurons from monkey DT). (**C**) Comparison of each neuron's activity between trials that saccades directed to the left or right target. Symbols represent the activity of single neurons. The horizontal and vertical bars represent the standard error of the mean (SEM) of pre-saccadic activity (-200~0 ms from saccade start). The filled dots denote that the activity is significantly different between left and right saccades (p<0.05, Wilcoxon test). (**D**) The correlation between perisaccadic activities (-150~100 ms after saccade end) of single PPS neurons (n = 55) and the end-position of the horizontal saccades. Each dot represents the correlation coefficient value of a single neuron. The histograms in horizontal and vertical axis represent the distribution of correlation coefficient and p value, respectively. (m: mean, p: p value, n: number of neurons).

The following figure supplements are available for figure 2:

**Figure supplement 1.** Correlation between single PPS neuron's pre-saccadic and post-saccadic activity.

**Figure supplement 2.** Correlation between the single PPS neuron's perisaccadic activity and the end-position of saccade when monkey made oblique saccades.

**Figure supplement 3.** The post-saccadic activity of PPS neurons is not evoked by foveal stimulation.

**Figure supplement 4.** PPS neurons show a similar firing pattern but different rate between tasks.

between trials to either the preferred or the non-preferred direction. Thus, the discharge difference between preferred and non-preferred direction (e.g., *Figure 2B–D*) cannot be due to the level of reward.

## The intended end-position was encoded by the relative change, rather than the absolute firing rate, of the perisaccadic activity of PPS neurons

Was the intended end-position signal task specific? We examined this question by comparing the activity of the same neuron in two different tasks. In both the SCS and MGS tasks, the monkeys made horizontal saccades to the same target locations. The firing patterns of PPS neurons were similar between two tasks, but the absolute firing rate differed between tasks (*Figure 2—figure*

*supplement 4A–B*). While the absolute firing rate was significantly higher in the SCS task than that in the MGS task (*Figure 2—figure supplement 4C*, in population level, p<0.001, paired t-test), the relative change in activity, comparing the perisaccadic interval (-150~100 ms of saccade end) with the interval of 100~300 ms before saccade onset, was very similar between two tasks (*Figure 2—figure supplement 4D*, in population level, p=0.2887, paired t-test). Such results suggested that the intended end-position was not task specific and it might be more reliably encoded by the relative change of activity of PPS neurons.

## The activity of LPS neurons reflects the actual end-position of saccade

In addition to the PPS neurons, we also recorded a group of neurons which started to discharge with a relatively long delay after saccade completion (LPS neurons, mean = 70.9 ms, STD = 31.3 ms). The activity of an example LPS neuron in the SCS task is shown in *Figure 3A*, and the population activity of 27 such neurons is shown in *Figure 3B*. The activity of LPS neurons was spatially tuned. Comparing the post-saccadic activities (25~225 ms after saccade end) of individual LPS neurons between trials in which saccades were directed to the opposite directions, all but two LPS neurons showed stronger activities in one direction than in the other (*Figure 3C*, p<0.05, Wilcoxon test). We then assessed the possibility that the activity of LPS neurons could consistently encode the actual end-

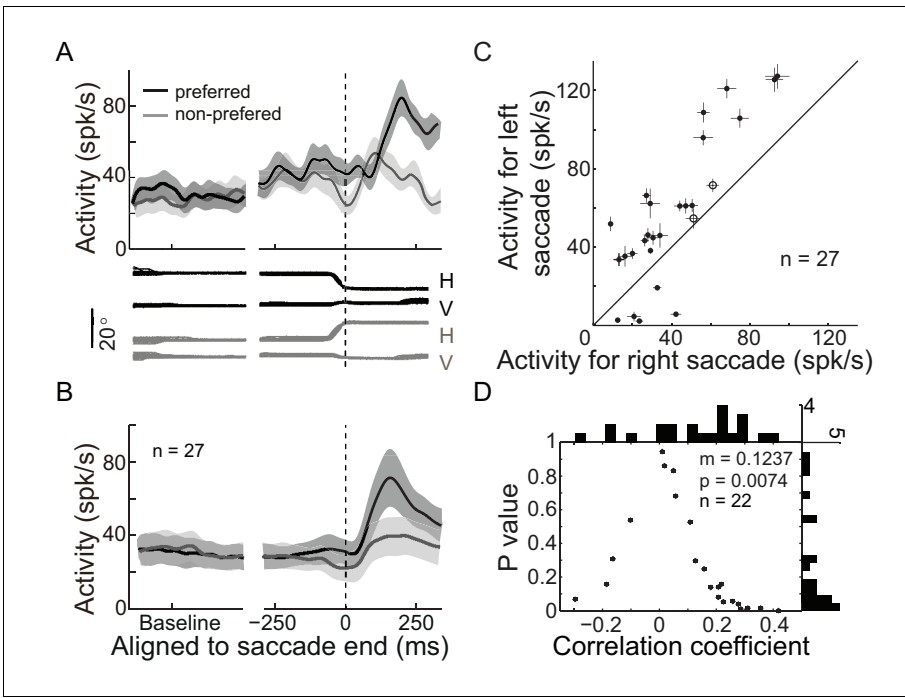

**Figure 3.** LPS neurons represent the actual end-position of saccades. (**A**) The activity of an example LPS neuron in the SCS task. Averaged spike density with 95% confidence interval (shaded area) in the preferred (black) and null direction (grey) are shown in the upper panel, whereas the horizontal and vertical eye traces are shown in the lower panel. (**B**) The averaged population activity of 27 LPS neurons. (**C**) Comparison of single neuron's activity between trials that saccades directed to the left or right target. Symbols represent the activity of signal neurons. The horizontal and vertical bars represent the standard error of the mean (SEM) of post-saccadic activity (25–225 ms after saccade onset). The filled dots denote that the activity is significantly different between left and right saccade (p<0.05, Wilcoxon test). (**D**) The correlation between postsaccadic activity (25~125 ms after saccade offset) of single LPS neurons (n = 22) with horizontal post-saccadic eye position when monkey made horizontal saccades. Each dot represents the correlation result of a single neuron. The horizontal and vertical histograms represent the distribution of correlation coefficient or p value, respectively.

The following figure supplement is available for figure 3:

**Figure supplement 1.** LPS neurons discharged similarly in different tasks.

position of saccades. If single LPS neurons code (at least partly) linearly for the end-position, one would expect a significant correlation of their activity with the random jitter in the actual saccadic end-position. If the code were consistent across single LPS neurons, we would expect a consistent bias of the correlation in one direction (negative or positive). Thus, we analyzed the correlation between the activities of LPS neurons and the end-position of saccades on a single-trial basis. 22 of 24 LPS neurons were recorded with enough trials (>30 trials in the preferred direction) when the monkeys performed the horizontal saccades. Indeed, we found that the distribution of the correlation coefficient values of these 22 LPS neurons was significantly biased to the positive direction (*Figure 3D*, mean r = 0.1237, p=0.0074, paired t-test). On a single neuron basis, we found that 18 out of 22 LPS neurons tended to be positively correlated with the end-position of saccades, with 5 reaching statistical significance (p<0.05). Taken together, although subject to noise on a single neuron basis, our results show that, in contrast to the PPS neurons, the activity of LPS neurons was positively correlated with the actual end-position of saccades.

Furthermore, unlike the PPS neurons, the LPS neurons showed similar post-saccadic activity between different tasks (*Figure 3—figure supplement 1A*: example neuron; *Figure 3—figure supplement 1B*: population neurons). Both the absolute firing rate (*Figure 3—figure supplement 1C*, in population, p=0.6526, paired t-test) and the relative change of the normalized post-saccadic activity (*Figure 3—figure supplement 1D*, in population, p=0.7262, paired t-test) were similar between the two tasks. Such task-independent activity further supports the notion that LPS neurons might encode the actual end-position of saccades.

## The PPS neurons behaved like a comparator to estimate the discrepancy between the intended and the actual end-positions of saccade

We first assessed the possibility that the activities of PPS and LPS neurons fitted the temporal request for the comparison between intended and executed action (*Figure 1A*). To do so, we superimposed the population activities of PPS and LPS neurons to directly compare their temporal characteristics (*Figure 4A*). We proposed the middle point of the activity change as an index to represent the temporal feature of neural activity. On average, the LPS neurons started to increase their activity 25 ms before the decay of PPS neurons, indicating the temporal overlap of the PPS and LPS signals in PPC.

Then, we analyzed whether and how the activities of the PPS and LPS neurons correlated with the monkeys' saccadic eye movements. We grouped the trials into different subsets based on the end-position of saccade in the horizontal meridian when the monkeys made horizontal saccades to the same target in the SCS task. The activities of three example subsets of trials are shown in *Figure 4B* for LPS neurons and in *Figure 4C* for PPS neurons. Notably, the activities of LPS neurons differed among subsets shortly after the increase in their activities (*Figure 4B*, within solid rectangle) and remained separated for at least 300 ms (about end of the trial). On the other hand, the perisaccadic activities of PPS neurons were similar among different subsets (*Figure 4C*, within dashed rectangle) but differed after the rise of the LPS neurons' activity (within solid rectangle).

To more systematically analyze the correlation between neuronal activities and the end-positions of saccade, trials were grouped into 6 subsets for LPS neurons and 16 subsets for PPS neurons, based on the horizontal end-position of saccade. The averaged post-saccadic activity (25~125 ms after saccade end) of the LPS neurons was positively correlated with the end-position of saccade (*Figure 4D*, red, r = 0.9791, p=0.0006, t-test). In contrast, the perisaccadic activity of PPS neurons (-150~100 ms to saccade end) did not vary among different end-positions of saccades (*Figure 4D*, black, r = -0.1355, p=0.6168, t-test). However, shortly after the arrival of the late post-saccadic activity of LPS neurons (150–350 ms after saccade end), the discharge level of the PPS neurons varied among different subsets of trials (*Figure 4C*, within solid rectangle); that is, the farther away the end-position of saccades is from the location of saccadic target, the higher the activity of PPS neurons.

Strikingly, in a late post-saccadic interval (150–350 ms after the saccade end), the normalized activities of PPS neurons were highly correlated with the end-position of saccade and showed a parabolic distribution (*Figure 4E*, black dots) with the lowest activity when saccades ended near the target location (10°). Data from hypometric (amplitude < 10°) and hypermetric (amplitude > 10°) saccades were nicely fitted with different regressions (*Figure 4E*, black lines). The correlation analysis

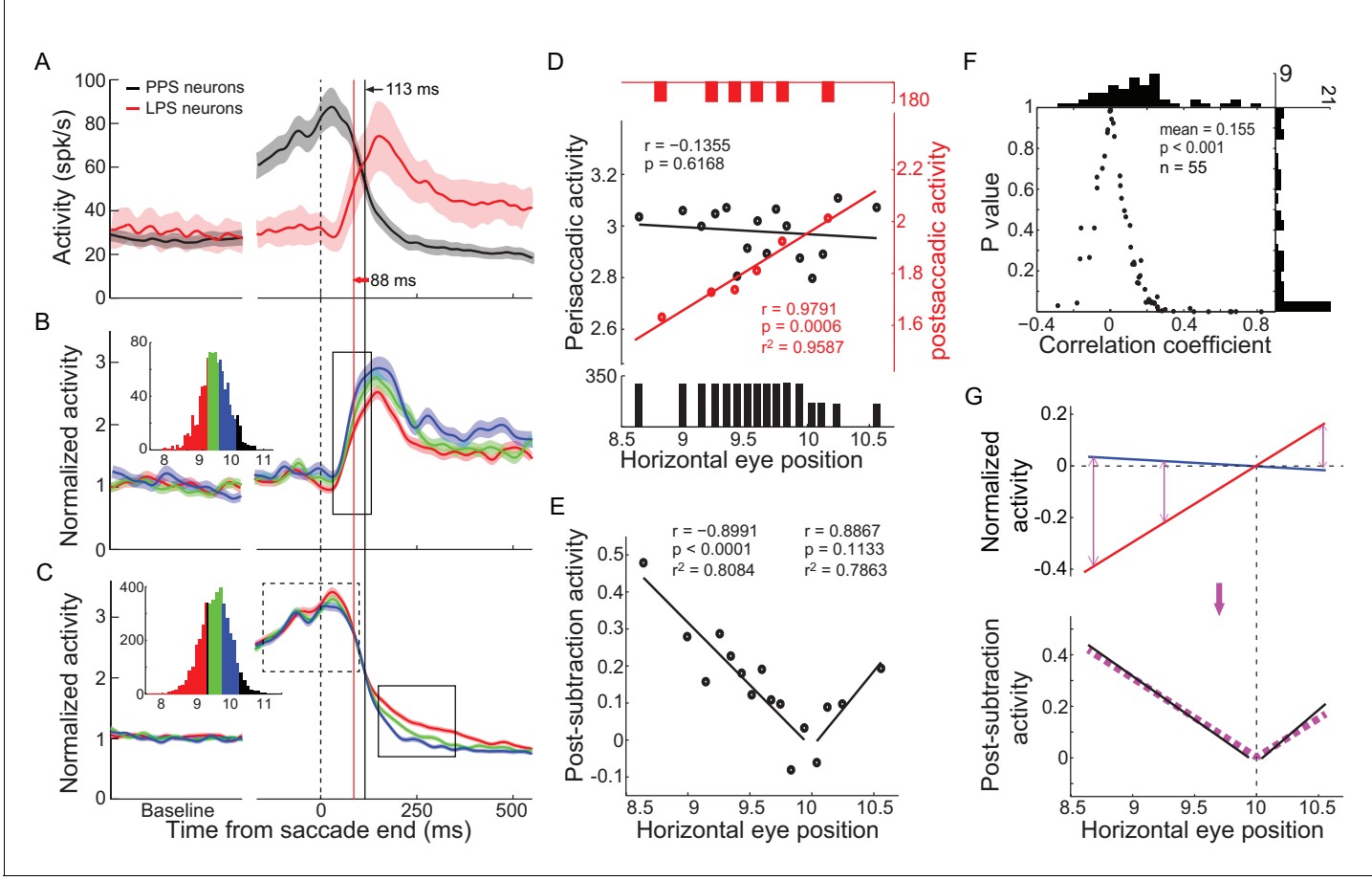

**Figure 4.** The integration between PPS and LPS neurons and the correlation with the end-position of saccades. (A) The temporal relationship between the activities of two types of neurons. The averaged activity with a 95% confidence interval of PPS neurons (black, n = 89) and LPS neurons (red, n = 27) was superimposed. The black and red vertical lines marked the time of the middle point of activity change for PPS neurons and LPS neurons, respectively. (B-C) The mean activity of LPS neurons (B) (n = 24) and PPS neurons (C) (n = 55) in three subsets of trials that were grouped based on the post-saccadic eye position in the horizontal meridian. The inserted histogram represents the distribution of post-saccadic eye position in the horizontal meridian. Colored curves represent the population activity of trials that have same color in the inserted histogram. The solid rectangle in (B) marks the postsaccadic period which was used for the analysis in (D), while the dashed and solid rectangles in (C) mark the perisaccadic internal used in (D) and post-subtraction interval used in (E), respectively. (D) The correlations of the late post-saccadic activity (25~125 ms after saccade offset) of LPS neurons (red) and perisaccadic activity (-150~100 ms relative to saccade offset) of PPS neurons (black) with post-saccadic eye position in the horizontal meridian. The colored dots represent the normalized activity of correlated trials for two types of neurons, respectively. The colored lines represent the regression linear fitting of correlated activity. The black and red bars in the bottom and top represent the trial number of each group. (E) The correlation between the post-subtraction activity (150~350 after saccade offset) of PPS neurons and the horizontal postsaccadic eye position. The larger the difference between the postsaccadic eye position and target location, the greater the post-subtracted activity in both undershoot and overshoot saccades. (F) The correlation between post-subtraction activity (150~350 ms after saccade offset) of single PPS neuron (n = 55) with saccadic error (distance between post-saccadic eye position and target position in horizontal meridian). Each dot represents the correlation result of a single neuron. The horizontal and vertical histogram represents the distribution of correlation coefficient or p value, respectively. (G) The post-subtraction activity fits well with the absolute difference between intended and real eye position signals. The blue and red lines represent the linear fitting of normalized perisaccadic activity of PPS neurons (blue, intended eye position signal) and post-saccadic activity of LPS neurons (red, actual end-position signal) with horizontal post-saccadic eye positions, respectively. The black lines represent the linear fitting of post-subtraction activity of PPS neurons with horizontal post-saccadic eye positions. The pink dashed line represents the regression fitting of the absolute subtraction result between the post-saccadic activity of LPS neurons and the perisaccadic response of PPS neurons.

The following figure supplements are available for figure 4:

**Figure supplement 1.** Averaged Pearson correlation coefficient (CC) analysis between post-subtraction activity of PPS neurons (n = 55) and saccadic errors for horizontal saccades in SCS task.

**Figure supplement 2.** Correlation between post-subtraction activity and oblique saccadic errors.

*Figure 4 continued*

**Figure supplement 3.** Correlation between post-subtraction activity and horizontal saccades in color-cue delayed saccades.

between single PPS neuron activity and the magnitude of saccadic error (difference between the saccadic end-position and the target position) further confirmed this result. While 12 out of 55 PPS neurons showed negative correlation (2 neurons reached statistically significant level), 43 out of 55 neurons showed positive correlation, and 19 neurons reached statistically significant level (p<0.05, t-test). Overall, the population correlation data showed a significant positive correlation with the magnitude of saccadic error (*Figure 4F*, mean r = 0.155, p<0.0001, paired t-test).

How did PPS neurons compare the intended and actual end-position signals? A simple mathematic model to estimate the congruence of two signals could be derived via subtraction. To make these two signals comparable, we set the activity in trials with saccades ending at target location (no saccadic error) as a reference condition for both PPS and LPS neurons. Then, for trials with saccades ending at other locations, their activities were normalized by subtracting the activity in the reference condition (*Figure 4G*, blue and red lines). Consistent with the subtraction model, the absolute difference between the two regression fittings of normalized activities of PPS and LPS neurons (*Figure 4G*, purple dashed lines) clearly overlapped with the activity of PPS neurons in the late post-saccadic interval (150–350 ms after saccade end) (*Figure 4G*, black lines). Moreover, the correlation between population activities of PPS neurons in the late post-saccadic interval and the magnitude of saccadic errors was significantly higher (p=0.0314, paired t-test) than the correlation with the end-position of saccade (mean r = 0.1134). It was also significantly higher (p=0.0023, paired t-test) than the correlation with the saccade amplitude (mean r = 0.0898). Thus, the level of the post-subtraction activity of PPS neurons was correlated positively with the discrepancy (error) between the intended (target location) and the actual end-position of saccade. However, the post-subtraction activity did not differentiate whether the error was from a hypometric or a hypermetric saccade.

To examine whether the chosen window (150–350 ms after the saccade end) for error presentation was merely coincidental, we analyzed the correlation between post-subtraction activity of PPS neurons and saccadic errors by using a 100 ms sliding window with a step of 10 ms. The sliding window started at 0 ms and stopped at 450 ms after saccade end. As shown in *Figure 4—figure supplement 1*, the population post-subtraction activity showed significant positive correlation with saccade error in a relatively long interval (120–380 ms after saccade end). In particular, during 150–300 ms the correlation was significantly positive (mean r > 0.1) and stable. Therefore, our results suggested that the PPS neurons might estimate the congruence between intended and actual end-position of saccades through subtracting the actual signal from intention signal.

A similar correlation between post-subtraction activity of PPS neurons and saccadic error was obtained in oblique trials of the SCS task (*Figure 1B*, with target 13° eccentricity) (*Figure 4—figure supplement 2A–C*) and in another task (CCS, *Figure 2C*, with target 10° eccentricity) (*Figure 4—figure supplement 3A–C*).

## The post-subtraction activity of PPS neurons was highly correlated with the probability of a secondary saccade

Finally, we examined whether the post-subtraction activity was correlated with the monkey's corrective behavior. In 1009 out of 4090 trials, the monkeys made a secondary (corrective) saccade following the primary saccade. The majority of the secondary saccades (851 of 1009) were directed toward the saccade target. We found that the probability of making a secondary saccade was reduced following the reduction of the primary saccadic error (*Figure 5A*). Also, the post-subtraction activity of PPS neurons was positively correlated with the probability of making a secondary saccade (*Figure 5B*, r = 0.8289, p=0.0001 t-test). Furthermore, when comparing trials with and without a secondary saccade, the post-subtraction activity was significantly higher in the trials with a secondary saccade (*Figure 5C* and *Figure 5D*). The positive correlation between the post-subtraction activity of PPS neurons and the probability of making a secondary saccade was also observed in the color-cue delayed saccade task (*Figure 4—figure supplement 1D–E*) and in the oblique saccade of the SCS task (*Figure 4—figure supplement 2D–E*).

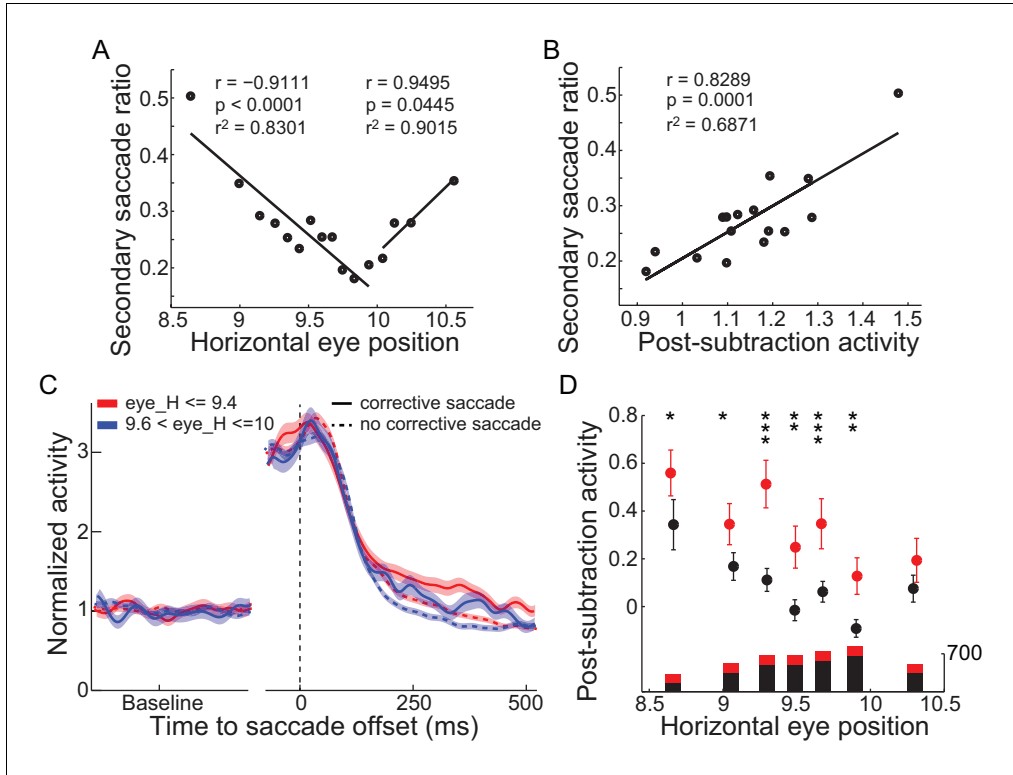

**Figure 5.** The correlation between the post-subtraction activity and the secondary saccades. (**A**) The correlation between the probability of making corrective saccades and the post-saccadic eye position. Data showed the lower the saccadic accuracy, the higher the secondary saccade ratio. Two black lines represent the linear fittings of undershoot and overshoot saccades, respectively. (**B**) Correlation between the post-subtraction activity and the occurrence of a secondary saccade. The data groups were the same as in panel (**A**). (**C**) The average activity in trials with and without corrective saccades. Among trials with similar horizontal post-saccadic eye position, the post-subtraction activity was higher in trials with secondary saccades (solid curves) than in trials without secondary saccades (dashed curves). (**D**) The comparison of the activity between trials with and without secondary saccades. All data sets show that trials with secondary saccades have higher post-subtraction activity. The length of the bars under each data point represents the trial number in each group (red: trials with corrective saccade; black: trials without corrective saccade). Asterisks indicate the following: *p<0.05, **p<0.01, ***p<0.001 (Wilcoxon test).

## The PPS and LPS neurons were intermixed in PPC

To examine how PPS and LPS neurons were distributed in PPC, we reconstructed the recording sites of individual neurons in a three-dimensional coordinate map for each monkey (*Figure 6*). Neurons were recorded from a 2 cm diameter recording chamber. The recording chamber was implanted under the stereotactic position, centered at 13 mm lateral to the middle sagittal line, and 3 mm posterior to the middle coronal line for monkey B; and at 13.5 mm lateral to the middle sagittal line and 4 mm posterior to the middle coronal line for monkey D. In the reconstructed map, the X and Y coordinates were relative to the center of the recording chamber of each monkey. Data showed that the distributions of the PPS and LPS neurons largely overlapped in both monkeys. Such intermingled distributions of PPS and LPS neurons indicated that PPS and LPS were recorded from the same area of PPC, suggesting the functional interaction between the two types of neurons is possible.

## Discussion

In the present study, we reported two groups of PPC neurons, which encoded the intended and the actual end-positions of saccade, respectively, and the difference between the two signals was highly correlated with the saccade error and with the possibility of making a secondary saccade.

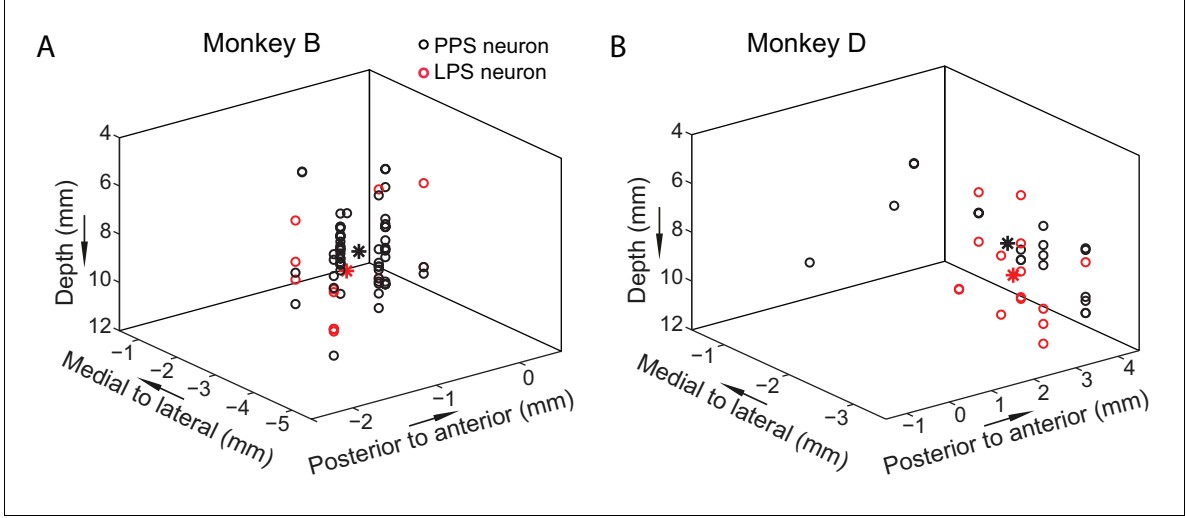

**Figure 6.** The 3D distribution of the PPS and LPS neurons. The distributions of two groups of neurons based on recording site are shown for both monkeys (**A**, **B**). Each circle represents one single neuron. The black and red stars mark the averaged center positions for PPS and LPS neurons, respectively. The mean distances between the PPS and LPS neurons are 1.78 mm and 1.0727 mm for monkey B and monkey D, respectively. The X and Y coordinates are relative to the center of the recording chamber.

In the experiment, the monkeys were trained intensively to make a saccade to a small target (0.2°) within a 3° checking window. The saccadic target remained visible in the SCS and CCS tasks (*Figure 1B, D*) while the monkeys made saccades. More importantly, the end-positions of saccades for rewarded trials were distributed closely around the target as a Gaussian-like distribution with STD = 0.49°. Thus, it is reasonable to assume that monkeys did intend to make saccades to the target location. Here, we found a group of PPC neurons that discharged persistently in both pre- and post-saccadic epochs (PPS neurons). Although this perisaccadic activity exhibited significant difference between left and right saccades (*Figure 2A–C*), it was not correlated with the actual end-position of saccade (*Figure 4D*, black). Therefore, we propose that the perisaccadic activity of PPS neurons represented the intended end-position of saccade (i.e., the location of the saccadic target) (*Steenrod et al., 2013*).

Another group of neurons is marked by late post saccadic activity (LPS). The activity of these neurons rises ~70 ms after saccade completion (*Figure 3B*). Such late post-saccadic activity unlikely represents the efference copy that emerges before the initiation of saccade (*Helmholtz, 1925*; *Sherrington, 1918*). In contrast, the temporal profile of this LPS activity is very similar to the extraocular proprioceptive response reported previously (*Prevosto et al., 2009*; *Wang et al., 2007*). Moreover, for the following reasons, we believe that the activity of LPS neurons represents more likely the extraocular proprioceptive signal rather than the visual signal. Firstly, the LPS neurons did not have visual responses in all three oculomotor tasks. Secondly, when aligning the activity of LPS neurons at 'feedback on' in the MGS task (*Figure 1C*), the activity increased even before the time of feedback stimulus onset. Thirdly, we previously reported that the extraocular proprioceptive signal rose up in the primary somatosensory cortex ~30 ms after the end of saccade (*Wang et al., 2007*). In our case, the LPS neurons started to discharge ~70 ms after saccade (*Figure 4A–B*), indicating that the LPS neurons might encode an extraocular proprioceptive signal (*Genovesio et al., 2007*).

More interestingly, we found that the PPS neurons in PPC behaved like a comparator that calculated the difference between the intended and actual end-position of saccade. The post-subtraction activity of PPS neurons was correlated highly with the magnitude of saccadic error and with the probability of making a secondary saccade. Considering the fact that PPC is not directly involved in the control of saccadic eye movement (*Li et al., 1999*; *Thier and Andersen, 1998*; *Wardak et al., 2002*), the saccadic error signal in PPC might function differently from the error signal that drives the premotor circuitry for saccade in the brainstem. One possible function may be the programming of a secondary saccade in case of the primary saccade missing the target. Our data show that the

comparison between the intended and the actual end-position of saccade starts in PPC ~70 ms after the end of the primary saccade (*Figure 4A–C*), which is too late to influence the primary saccade but is early enough to control the secondary saccade (the latter has a latency of ~154 ms in average). Therefore, this parietal error signal may reflect the next of programming a secondary saccade, a possibility that is suggested by the correlation between secondary saccade probability and the post-subtraction activity of PPS neurons. In contrast, the brainstem-cerebellum models do not explain how a secondary saccade is planned.

The widely studied short-term saccadic adaptation may be fully explained by the brainstem-cerebellum machinery. Nevertheless, the cortical error representation can still be useful, for instance, for certain features of adaptation that go beyond the standard short-term adaptation. In parallel to short-term adaptation, a long-term adaptation (i.e., the post-adaptation effect lasting for days) is present (*Barash et al., 1999*; *Robinson et al., 2006*), and the neural basis for the long-term adaptations remains unclear. There is evidence for meta-learning, that is, short-term adaptation becoming more rapid with repeated training (*Kojima et al., 2004*; *Wong and Shelhamer, 2011*); there are even contentions of 'cognitive' modulation of short-term saccadic adaptation (*Cotti et al., 2009*; *Srimal and Curtis, 2010*). The cortical error representation observed in the present study may have to do with any of these processes.

Another phenomenon is that a saccade can be elicited either by the sudden appearance of an external stimulus (reactive saccade) or by an endogenous cue (voluntary saccade). The cumulative evidence has shown that the saccadic adaptation of reactive and voluntary saccades involves different mechanisms and partially separated neural circuitries (*Cotti et al., 2009*; *Deubel, 1995*; *Erkelens and Hulleman, 1993*; *Gaveau et al., 2005*; *Hopp and Fuchs, 2010*; *Panouilleres et al., 2014*; *Zimmermann and Lappe, 2009*). While the brainstem-cerebellum circuitry plays a crucial role in the adaptation of reactive saccades (*Barash et al., 1999*; *Hopp and Fuchs, 2004*; *Optican and Robinson, 1980*; *Panouilleres et al., 2015*; *Robinson et al., 2002*; *Takagi et al., 1998*), PPC is highly involved in the adaptation of voluntary saccade (*Gerardin et al., 2012*; *Panouilleres et al., 2014*). The parietal error signal found in the present study might contribute to the latter.

Recent studies have shown that comparison between motor intention and proprioception is critical for motor learning (*Desmurget and Grafton, 2000*; *Panouilleres et al., 2015*; *Wolpert et al., 2011*; *Wolpert and Ghahramani, 2000*) and motor-related cognitive functions, such as motor awareness (*Berti and Pia, 2006*; *Fink et al., 1999*; *Haggard, 2005*) and self-recognition (*Haggard, 2008*; *Jeannerod, 2003*; *Knoblich, 2002*; *van den Bos and Jeannerod, 2002*). A line of evidence suggests that PPC of primates plays an important role in the comparison between motor intention and proprioception (*Gerardin et al., 2012*; *Panouilleres et al., 2015*). Clinical studies also found that patients with lesions in PPC frequently exhibit difficulties in evaluating and comparing the intended and actual positions for awareness of self-movement (*Haarmeier et al., 1997*; *Sirigu et al., 1999*), for maintaining and updating an internal representation of action state (*Kawato and Wolpert, 1998*), and for motor error correction (*Pisella et al., 2000*). Here, our study adds the neuronal evidence for motor error computation in PPC.

## Materials and methods

### Animal preparation

Two male rhesus monkeys (6–8 kg, 6–7 years old) participated in the present study. They were housed in separate cages in a large room with a 12 hr light/dark cycle. The horizontal and vertical eye position signals were recorded using the scleral eye coil technique (Crist Instrument Sclera), and data were sampled at 1 kHz. Before training, each monkey was surgically implanted with a head post and two eye coils. After training in all three oculomotor tasks, a recording chamber was implanted above PPC of the right hemisphere for chronic electrophysiological recording. All experimental and surgical procedures were standard and approved by the Animal Care Committee of Shanghai Institutes for Biological Sciences, Chinese Academy of Sciences; Animal Care and Ethics Committee of Beijing Normal University.

## Behavior tasks

Visual stimuli were projected (View sonic, PJD7383) onto a large screen that was placed 80 cm in front of the monkeys' eyes. We used a QNX computer to control the visual display and to run the real-time data acquisition system (REX; NIH, Bethesda, MD).

### Spatial-cue delayed saccade task (SCS, *Figure 1B*)

There were two versions of the SCS task: horizontal (with target at 10° eccentricity) and oblique (with target at 13° eccentricity) directions of saccadic targets. During training and data collection, the two versions were presented in separate sessions. Trials began with a central fixation point (FP) that appeared in the center of the screen. The monkeys were required to keep their fixation on this point for as long as it was on the screen. Next, 500 ms after the monkey achieved central fixation, two identical dots (targets) appeared simultaneously, one on the left and one on the right of the fixation point. These two dots remained visible until the end of the trial. 400 ms after the appearance of the two dots, a visual cue randomly flashed for 200 ms in one of 24 locations on the screen. Another 400 ms after the visual cue offset, the central fixation point disappeared. The monkeys needed to make a saccade to the dot that was on the same side of the visual field as the visual cue appeared.

During the daily training, monkeys were always encouraged to make a saccade to the target as accurately as possible. We used a small check window (3°) to check monkeys' post saccadic eye position for every trial. In addition, the saccadic target was very small (0.2°) in order to avoid a wide variation of the saccade end points. In fact, the end points of saccades in both monkeys showed narrow Gaussian-like distribution that was centered near the target location (mean = 9.6615 degree, STD = 0.4894 degree). Thus, our monkeys were striving for the exact position of the target in the tasks.

### Memory-guided saccade task (MGS, *Figure 1C*)

Trials began with the appearance of a fixation point (FP) at the center of the screen. Monkeys needed to fixate on the FP for as long as it was on. After 500 ms, a visual target briefly appeared (200 ms) at one of eight potential locations that were evenly spaced and positioned at equal eccentricity (10°). Monkeys had to maintain central fixation until the fixation point offset and then had to make a single saccade toward the memorized target location. Afterwards, a visual stimulus appeared in the same location of the target as the visual feedback.

### Color-cue delayed saccade task (CCS, *Figure 1D*)

Trials began with the appearance of a fixation point (FP) at the center of the screen. Monkeys needed to fixate on FP for as long as it was on. After 500 ms, two visual stimuli with different colors (one red and one green) appeared simultaneously, and 600 ms later, the fixation point disappeared and monkeys needed to make a saccade to the red stimulus.

## Neuronal recording

We used a micromanipulator (NAN Instruments) to drive the glass-covered tungsten microelectrodes (~1 MΩ), guided by a gauged stainless steel guide tube, down into the cortex to record activity in single neurons. Neural signals were conventionally amplified (Alpha Omega MCP Plus8) and then filtered through 300~3000 HZ (Krohn-Hite Model 3384). We used a QNX computer to run a real-time neuronal analysis system (MEX; NIH, Bethesda, MD) to sort the spiking signals based on the amplitude and waveform of spikes.

## Behavioral data analysis

Criteria for saccades: The start and end of a saccade were determined using a threshold for velocity- and template-matching criteria. The start of a saccade was defined as the time when the velocity exceeded 30°/s, and the end of a saccade was defined as the point at which the velocity became less than 10% of the peak velocity. Furthermore, trials were only included for further analysis if they fitted the following criteria: (1) the saccade duration was 10 to 100 ms; (2) the saccadic endpoint was within a 5° window that was centered at the saccade target; (3) the saccadic amplitude was larger than 4°; and (4) the saccadic latency was shorter than 500 ms. Furthermore, we set different criteria for secondary saccade analysis: (1) the saccade duration was longer than 5 ms; (2) the minimum velocity for saccade onset was 8°/s; (3) the saccadic amplitude was larger than 0.2°; and (4) the

secondary saccade occurred at least 100 ms after the end of the first large saccade. It has been reported that in human subjects, up to 20% of secondary saccades were not directed to the saccade target. These secondary saccades were also considered as secondary saccades (*Morel et al., 2011*). Therefore, we included all secondary saccades for further analysis. Furthermore, we defined the average eye position during 0–20 ms after the saccade offset as the saccade end point.

## Neuronal data analysis

### Classification of recorded neurons

We defined a neuron as a pre- and post-saccadic response neuron if its activity fitted the following criteria: (1) the mean activity during the pre-saccadic interval (0–200 ms before the saccade onset) in the preferred direction was significantly greater than the mean activity in the baseline interval (0~500 ms after fixation onset) ($p<0.01$, paired t-test); (2) at least 5 out of 10 bins (bin width = 20 ms) during the pre-saccadic interval had significantly higher activity than the mean activity in the baseline interval ($p<0.05$, paired t-test); (3) the mean activity during the pre-saccadic interval in the preferred direction was significantly greater than the mean activity during the pre-saccadic interval in the non-preferred direction ($p<0.01$, Wilcoxon test); and (4) the decay of activity was later than 35 ms after the saccade completion. In total, we isolated 380 neurons that significantly increased in activity before saccade initiation in the SCS task. Among them, 89 neurons were classified as pre- and post-saccadic response neurons.

We defined the late post-saccadic response neurons according to the following criteria: (1) the mean activity during the pre-saccadic interval was not significantly different from the mean activity in the baseline interval ($p>0.05$, paired t test); (2) the mean activity from 150 ms before the saccade completion was neither significantly different from the mean activity at baseline nor the mean activity in the non-preferred direction ($p>0.05$, Wilcoxon test); (3) the mean activity in one of two post-saccadic intervals (0~200 ms or 100~300 ms after saccade completion) was significantly greater than the mean activity at the baseline and the mean post-saccadic activity in the non-preferred direction ($p<0.05$, Wilcoxon test); (4) in at least 2 of 4 bins (bin width = 50 ms), the post-saccadic activity was significantly greater than that at the baseline interval; and (5) the increase in the time of post-saccadic activity was longer than 35 ms after the saccade completion. In the SCS task, 110 neurons exhibited significant post-saccadic activity. Among them, 27 were classified as late post-saccadic response neurons.

Data shown in *Figure 2* and *Figure 2—figure supplement 1–2* and *Figure 4—figure supplement 1* includes all neurons that fitted the above criteria. Data shown in *Figure 4C–H* and *Figure 5* includes neurons with enough trials (>30 trials) in the preferred direction (55 out of 62 for pre- and post-saccadic response neurons, 19 out of 24 for late post-saccadic response neurons). Also, data shown in *Figure 4—figure supplement 1–2* includes neurons with enough trials in the preferred direction (in *Figure 4—figure supplement 2*, trials > 20; in *Figure 4—figure supplement 3*, trials > 30).

### Method for determining the decay of pre-saccadic activity

Because the pre-saccadic activity decreased exponentially mostly around the time of the saccade, we employed an exponential function to fit the mean activity of each neuron to determine the optimal decay time. Specifically, we used the following method: (1) Calculating the mean spike density of each neuron in the preferred direction using a Gaussian kernel function ($\sigma$ = 5 ms) and then selecting a time interval (200 ms before the saccade onset to 300 ms after the saccade onset, alignment at saccade onset; 250 ms before saccade completion to 250 ms after saccade completion, alignment at the saccade completion) for further analysis; (2) Using a small moving window (bin width = 20 ms, step = 1 ms) to determine the peak activity point, which was selected as the start point for the exponential fitting; (3) The end point for the exponential fitting was defined as the lowest activity after the peak activity point; (4) Because neurons discharged with large variations, we defined a period of peak activity fluctuation, which contained the activity decay time. This period ran from the peak activity point to a point after which the activity was lower than the peak activity by 2 STE, or 80% of peak activity; (5) Using an exponential function ($y = A(1)*e^{(x/A(2))}+A(3)$) to fit the spike density function starting from each time point within the fluctuation period to the end point. The start point of the best fit, which had the minimum mean square residual error, was the activity decay time.

## Method for determining the increase in post-saccadic activity

We also used the exponential fitting method to determine the optimal increase in the time of post-saccadic activity: (1) using the same method as mentioned in the previous section to calculate the mean spike density function in the preferred direction, selecting the time interval for further analysis, and finding the peak activity; (2) finding the bin with the lowest activity before the peak point; (3) defining the fluctuation period of lowest point as the lowest activity point to the last bin in which the activity was lower than 2 STE above the mean activity of the lowest activity or 2 STE above the mean activity of baseline; (4) the start point with the best exponential fit was the increasing time points of post-saccadic activity.

## Statistic functions used in analyzing neuronal activity

Two standard statistic functions were used in the present study (t-test and Wilcoxon test). The normalized single neuron's activity was calculated by using the baseline activity (100∼500 ms after fixation point onset) to divide the activity throughout the trial.

## Choosing different temporal windows to analyze different signals

We chose a time window to analyze a signal based on its biological meaning and the model framework. For instance, the actual saccade end-position signal (proprioceptive input) usually reaches the cortex with a delay of ∼30 ms after a saccade completion (*Wang et al., 2007*; *Xu et al., 2012*). Since the monkeys might make secondary saccades at ∼250 ms after the primary saccades, we chose a window, 25–125 ms after saccade completion, to study the actual saccade end-position signal. The intended saccade end-position signal should appear before saccade initiation and last until the arrival of the actual eye position signal. Thus we chose a time window from 150 ms before to 100 ms after the saccade end to study the intentional signal. Finally, the error signal should be generated after the comparison of internal and external signals, we chose a window (150–350ms after the saccade completion) for error signal analysis. Although varying the time interval may quantitatively change the results slightly, it will not change the major conclusions qualitatively.

## Acknowledgements

We thank Prof. Michael E Goldberg and Dr. Malte Rasch for their helpful suggestions and comments. We thank IDG/ McGovern Institute for Brain Research at Beijing Normal University for the grant support.

## Additional information

### Funding

| Funder | Grant reference number | Author |
|---|---|---|
| Ministry of Science and Technology of the People's Republic of China | 2011CBA00406 | Mingsha Zhang |
| National Natural Science Foundation of China | 31471069 | Mingsha Zhang |
| National Natural Science Foundation of China | 91432109 | Mingsha Zhang |

The funders had no role in study design, data collection and interpretation, or the decision to submit the work for publication.

### Author contributions

YZ, Designed the experiments, Trained monkeys and collected the behavioral and neuronal data, Ana; YL, Trained monkeys and collected the behavioral and neuronal data, Conception and design, Analysis and interpretation of data; HL, Drafting or revising the article, Contributed unpublished essential data or reagents; SW, Conception and design, Analysis and interpretation of data, Drafting

or revising the article; MZ, Designed the experiments, Supervised the experiments, Analysis and interpretation of data, Drafting or revising the article

## Author ORCIDs

Mingsha Zhang, http://orcid.org/0000-0002-5407-7770

## Ethics

Animal experimentation: Two male rhesus monkeys (6-8 kg, 6-7 years old) were involved in the present study. They were housed in separate cages in a large room with 12 hours light/dark cycle. The horizontal and vertical eye positions signals were recorded using the scleral eye coil technique (Crist Instrument Sclera), and data were sampled at 1 kHz. Before training, each monkey was surgically implanted with a head post and two eye coils. After training in three oculomotor tasks, a recording chamber was implanted above the posterior parietal cortex of the right hemisphere for chronic electrophysiological recording. All experimental and surgical procedures were standard and approved by the Animal Care Committee of Shanghai Institutes for Biological Sciences, Chinese Academy of Sciences (Project number ER-SIBS-221112P); Animal Care and Ethics Committee of Beijing Normal University. (Project number IACUC (BNU) - NKLCNL 2013-09)

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
