## [Decision Letter]

[Editors’ note: a previous version of this study was rejected after peer review, but the authors submitted for reconsideration. The previous decision letter after peer review is shown below.]

Thank you for choosing to send your work entitled "Neural representation of the computation of motor error by single neurons in the parietal cortex" for consideration at *eLife*. Your full submission has been evaluated by Eve Marder (Senior editor), a Reviewing editor, and three peer reviewers, and the decision was reached after discussions between the reviewers. Based on our discussions and the individual reviews below, we regret to inform you that your work will not be considered further for publication in *eLife* at this point. However, if you would be able to address all issues raised by the reviewers in a satisfactory manner, we might consider a resubmission, although we cannot guarantee renewed refereeing by the same or other reviewers nor ultimate acceptance of your manuscript.

We identified the following main problems (for details and other issues, please see the appended reviews):

1) There were major problems with the design of the study (explicit control of errors, reviewer 1). While this cannot be changed at this point, it would be mandatory to reevaluate the interpretation of these data.

2) There were major issues in the style and clarity of the text, including the description of hypotheses, and the English (all three reviewers).

3) There were major problems with data analysis and interpretation. These include, but are not limited to, the distinction of different neuronal types in LIP which was felt to be particularly important (reviewer 3).

*Reviewer #1:*

This manuscript addresses an important issue in motor control, evaluating at the neuronal level for the presence and integration of prediction and feedback signals in LIP neurons during saccades. These signals are critical to the operation of a forward internal model and computing sensory prediction errors. The study uses three types of delayed saccade paradigms, spatial-cue delayed (SCS), memory-guided saccades (MGS) and color-cue delayed saccades (CCS), focusing on the relation between neurons firing both before and after the saccade and neurons that fire after the saccade.

While interesting, the manuscript has major problems with the presentation, analysis and interpretations. The result leaves a reviewer confused and uncertain exactly what was done, the results and their implications. First is the paradigm. The critical arguments need to be based on the undershooting or overshooting of the target. However, the paradigm was not designed to produce these events, instead the study attempts to obtain that information based on the natural variation in saccade end point position. The analysis assumes that the monkey is always striving for the exact center of the target and, therefore the presaccadic firing does not vary with position but this is a big assumption and needs evidence to support. Without this implicit assumption, the critical analyses in Figure 5 break down. The study would be more convincing if position errors were explicitly controlled.

The second major problem is the presentation. The study ignores the cerebellum and cerebellum-like literature on sensory prediction error processing. These include the many psychophysical, imaging and patient studies showing that the cerebellum is critical to sensory prediction errors by Shadmehr, Krakauer, and Diedrichsen (see papers such (Diedrichsen et al., 2005; Izawa et al., 2012; Krakauer and Shadmehr, 2006; Shadmehr et al., 2010) (Tseng et al., 2007) as well as single cell studies by Sawtell (Kennedy et al., 2014; Requarth et al., 2014) and Ebner (Popa et al., 2012; Popa et al., 2014). Also the presentation of the Results is dominated by description and discussing of supplementary figures. This is somewhat distracting, in some cases the claims are not clear or the supplementary results are not that crucial. In the Methods, three paradigms were used. However, no data is presented from the color-cue delayed saccade paradigm.

The third major problem is the analyses and problematic use of statistics. Many claims are based on one-sided t-tests. The vast majority of statisticians agree that one-sided tests should only be used if the alternative is physically impossible. This is not the case here and should not be used. Also, the use of 95% confidence intervals as a measure of population variability is not an appropriate measure nor is SEM a good measure. Instead, standard deviation is a much more appropriate measure of variability and should be used. Also, the single cell analyses in Figure 5 that relate p-values to the correlation coefficients are problematic. First, obviously p-values and the correlation coefficient are highly related and will have somewhat of a bell-shaped profile. This is not informative. Second, performing secondary level statistics on correlation coefficients is not good statistical practice.

Fourth, there are critical claims in the paper that do not agree with the data presented. In Figure 5, the argument is made that the late post-saccadic activity is linearly related to horizontal eye position based on averaging and binning of the data. It is then claimed in Figure 5 that the same is true for individual cell firing (subsection “The pre- and postsaccadic response neurons behave like a subtraction operator that measure the discrepancy between predicted and actual eye position signals”, second paragraph). However, Figure 5 shows that the p-value for the vast majority of the cells is > 0.05. At best, 2 or 3 cells had a correlation coefficient that reached statistical significance. Furthermore, the correlation coefficients are equally distributed between positive and negative for the included cells in Figure 5, yet there is an overall positive relationship between post-saccadic activity and position in Figure 5. These discrepancies seem very hard to reconcile.

References:

Diedrichsen J, Hashambhoy Y, Rane T, Shadmehr R (2005) Neural correlates of reach errors. J Neurosci 25:9919-9931.

Izawa J, Criscimagna-Hemminger SE, Shadmehr R (2012) Cerebellar contributions to reach adaptation and learning sensory consequences of action. J Neurosci 32:4230-4239.

Kennedy A, Wayne G, Kaifosh P, Alvina K, Abbott LF, Sawtell NB (2014) A temporal basis for predicting the sensory consequences of motor commands in an electric fish. Nat Neurosci.

Krakauer JW, Shadmehr R (2006) Consolidation of motor memory. Trends Neurosci 29:58-64.

Popa LS, Hewitt AL, Ebner TJ (2014) The cerebellum for jocks and nerds alike. Front Syst Neurosci 8:1-13.

Popa LS, Hewitt AL, Ebner TJ (2012) Predictive and feedback performance errors are signaled in the simple spike discharge of individual Purkinje cells. J Neurosci 32:15345-15358.

Requarth T, Kaifosh P, Sawtell NB (2014) A role for mixed corollary discharge and proprioceptive signals in predicting the sensory consequences of movements. J Neurosci 34:16103-16116.

Shadmehr R, Smith MA, Krakauer JW (2010) Error correction, sensory prediction, and adaptation in motor control. Annu Rev Neurosci 33:89-108.

Tseng YW, Diedrichsen J, Krakauer JW, Shadmehr R, Bastian AJ (2007) Sensory prediction errors drive cerebellum-dependent adaptation of reaching. J Neurophysiol 98:54-62.

*Reviewer #2:*

This manuscript addresses the question of how the error between predictive and actual motor signals are represented in LIP. They propose that one population of neurons 'pre- and post-saccadic neurons' encode the desired target location and another the 'postsaccadic neurons' encode the actual target location, and that the difference between the signals of these two populations encodes the error signal. They also propose that larger error signal correlates with greater probability of corrective saccades.

The hypothesis is well described in Figure 1 and the experimental methods are well executed. However, the writing of the manuscript leaves much to be desired. It is often not clear what question is being addressed. Figures are described but what hypothesis is being addressed is left to be inferred by the reader; it would be extremely helpful to first state the hypothesis or prediction at the beginning of each section/paragraph and then describe the data. One important shortcoming is the lack of clarity regarding how the subtracted responses are computed; this is essential for the reader to understand the central tenet of this manuscript. I suspect that all this could be fixed with the help of an experienced writer. As it is, I found it difficult to read and therefore difficult to evaluate. However, if the writing is clarified and the data support the claims made, then it could be a highly impactful paper.

Comments:

Many English usage errors. e.g. Introduction, first paragraph: ‘…challenged by evidence that indicates…’, subsection “The activity of pre- and postsaccadic response neurons reflects the internal signals to predict the required future eye position”, end of first paragraph: should be “…postsaccadic activity decayed…”? At the start of the second paragraph of the aforementioned subsection: impending?

In the subsection “The activity of pre- and postsaccadic response neurons reflects the internal signals to predict the required future eye position”, second paragraph: 'in one direction than the other'. Do you mean in the preferred vs. non-preferred direction? Or do you mean in the left direction in the left saccade trials? Figure 3: are all of these 89 from left saccade trials?

In the subsection “The activity of pre- and postsaccadic response neurons reflects the internal signals to predict the required future eye position”, second paragraph: unclear what the point of this paragraph is. Is it trying to say that pre and postsaccadic activity typically correlated so both could be relevant signal?

In the subsection “The activity of pre- and postsaccadic response neurons reflects the internal signals to predict the required future eye position”, third paragraph: not clear why you are comparing postsaccadic SCS with foveal visual response in MGS. Is this the rationale? "One consideration we excluded is the possibility that the postsaccadic activity is induced by visual stimulation. If so, then one would expect to see response to foveal visual response in the MGS task." However, this rationale is still not very strong. One could have weak visual response that is potentiated more in the SCS task than MGS task.

In the subsection “The activity of pre- and postsaccadic response neurons reflects the internal signals to predict the required future eye position”, third paragraph: 'post-subtracted activity': what exactly is being subtracted from what? It would be helpful to delineated what portions of response from which neurons fit into the hypothesis outlined in Figure 1.

This is really difficult to parse: 'between the perisaccadic activity of the pre- and post-saccadic neurons and postsaccadic activity of the late postsaccadic response neurons'. Perhaps it would be helpful to shorten 'pre- and post-saccadic neurons' to something like PPS neurons.

It is not clear what the color cue task was used for.

*Reviewer #3:*

The manuscript at question addresses the role of efference copy (=corollary discharge) and proprioceptive information in movement control. The model system considered is visually guided saccades and the hypothesis, the authors present is that parietal area LIP has access to both efference copy information and proprioceptive feedback and moreover, that a particular class of neurons, dubbed presaccadic-postsaccadic (=perisaccadic) response neurons represents the difference between the two. This difference is thought to serve as an error signal, responsible for subsequent corrective movements.

This argument is based on the distinction of two pools of neurons found in areas LIP of rhesus monkeys. One pool involves neurons which start to fire after a saccade, assumed to represent the acquired eye position after a saccade, reflecting proprioceptive feedback. The second pool consists of neurons with perisaccadic responses thought to represent a prediction of the eye position. The major result is that the late activity of perisaccadic neurons seems to correspond to the difference between their early (presaccadic) response and the discharge of the postsaccadic neurons. Moreover, this difference measure and the late perisaccadic responses thought to represent the difference show an intriguing dependence on the final eye position. The dependence shows a minimum for the ideal position (i.e., for an eye position corresponding to target position) and increases with increasing deviation from this ideal in both directions. This result is intriguing! The dependence found is the one of an error representation suitable to drive corrections and eventually learning. Actually, it seems too specific to be an adventitious consequence of the many assumptions the authors have to make on their way to this particular result, assumptions whose biological significance one may question. Nevertheless, I am not fully convinced that the conclusions drawn are justified and, moreover, that the conceptual framework presented is viable. What are my concerns?

Differentiation of 2 groups of neurons: The authors separate 2 groups basically only based on differences in the amount of presaccadic activity. Yet, this is not sufficient to conclude that neurons falling on either side of the cut off chosen are qualitatively different. The problem associated with this approach can be easily demonstrated by considering the distribution of saccade errors discussed in Figure 5 distribution which is parsed by the authors into a number of error classes. Here they could as well have concluded that saccades found in distinct error classes are qualitatively different. Yet, they assume a continuous distribution. What is needed in order to justify the assumption of separate populations is a rigorous statistical approach (e.g., a cluster analysis), based on many more parameters.

Non-visual vs. visual basis of responses: independent of the question if the assumption of two neuronal pools is justified or not, I doubt that the interpretation of presaccadic activity is really as unambiguous as claimed by the authors. For instance, the neuron shown in Figure 3—figure supplement 2 shows a strong build-up of presaccadic activity with a clear peak at the time of the saccade when tested in the SCS paradigm but very little in the MGS paradigm, although probably in both tasks saccades having similar metric were carried out. The major difference between the two is the fact that in the SCS, but not in the MGS paradigm, the saccade target was available all the time. In other words, I would assume that the peripheral visual cue (based on the target) must have contributed significantly to the response in the SCS task. Independent of this specific interpretation: why should a response component, which – according to the authors – reflects the prediction of the saccade-based change in eye position should differentiate between paradigms? Further: why should an efference copy related discharge show the conspicuous build up plus peak associated with the saccade? And why should it begin such a long time before the saccade? My admittedly very subjective intuition is that such profiles reflect an intention and not an efference copy.

Tuning for eye position (change): If the presaccadic response component of the perisaccadic neurons and the responses of postsaccadic neurons reflected the predicted and experienced saccades, one would expect to see a clear reflection of saccade metrics. The paper does not provide any pertinent information: Do the authors find a tuning for the amplitude of the saccade vector and/or eye position? I think that a clear tuning must be demanded. Moreover, one would like to see that postsaccadic response neurons can be activated by passive eye movements.

Response windows: in order to quantify responses and the subtraction measure, the authors define windows without providing any justification for their specific choices. As they assume that the activity of perisaccadic neurons in the late window reflects the subtraction, nature would have to choose the windows like the authors unless the result were independent of the exact choice of windows. This question relates to the more general one of wiring: is there any biological basis for supporting the assumption that the output of postsaccadic neurons could be subtracted from the discharge of perisaccadic neurons? For instance, could it be that postsaccadic neurons are actually interneurons, potentially revealed by different spike waveforms and discharge statistics?

Conceptual concerns: I think that the conceptual framework presented in the Introduction and the Discussion suffers from a lack of clarity and rigor as to the role of parietal cortex in comparing an efference copy and actual eye position. I would guess that probably any oculomotor physiologist believes in the one or the other variant of Robinson saccade model in which the predicted momentary eye position is compared with the desired endpoint position and the difference is driving the saccade. Furthermore, the prediction is continuously updated based on error information, probably by cerebellar signals. This is a brainstem model with the superior colliculus representing the highest level. The neurons representing the prediction (i.e., the efference copy) in this model are the PPRF short-lead burst neurons whose discharge is precisely correlated with saccade kinematics, with a precision of msec. This model works perfectly without involving cortex, which would only disturb because of the unpleasant delays it would contribute. What I am trying to say is that in order to drive saccades, there is no need for the comparison of efference copy and actual eye position at the level of LIP. Such a comparison may take place, yet, the raison d´etre (e.g. perceptual purposes?) the authors have in mind should be presented much more clearly. As yet, I do not see a compelling concept and I do not see a need for control purposes.

[Editors’ note: what now follows is the decision letter after the authors submitted for further consideration.]

Thank you for resubmitting your work entitled "Neuronal representation of the motor errors in macaque posterior parietal cortex" for further consideration at *eLife*. Your revised article has been favorably evaluated by Eve Marder (Senior editor), a Reviewing editor, and three reviewers. The manuscript has been improved but there are some remaining issues that need to be addressed before acceptance, as outlined below:

We apologize for the delay in this decision, which was caused by extensive discussions among the editors and reviewers. Basically, one of the reviewers is strongly supportive of publication, one more negative, and the last midway between. Consequently, there are some remaining concerns that you will need to address in the text before a final decision can be made.

1) If evidence is obtained without a prior working hypothesis – one should at least try to discuss the findings in relation to dominating concepts a posteriori and try to argue why the latter may be insufficient, wrong or whatever. However, to use the standard model to justify the experiments is weak. Please rework the text with this perspective in mind.

2) You assume a specific subtractive mechanism. This requires a specific anatomical relationship and it implies continuous interactions. However, there is no evidence that the strong anatomical assumption is met. You searched in large parts of posterior parietal cortex, ignoring well-defined anatomical boundaries. Secondly, the assumption of error representation is based on the selection of rather arbitrary time windows. Please address in the Discussion whether the assumed error representation could be a serendipitous finding consolidated by later work. However, it might well be an artifact of the many non-substantiated assumptions made.

Specific comments for your attention:

It is still somewhat of a challenge to get at the robustness of the data. For example, in Figure 5, 18 out of 55 PPS neurons showed a significant position correlation with saccadic error and the population ρ is 0.13. Therefore, less than 2% of the firing variability is error-related. Is this sufficient encoding for the argument? Another example is in Figure 4 in which the correlation between post-saccadic activity of single LPS neurons and eye position is 0.10, or 0.01% of the variability. However, that is an issue best decided by the wider scientific community.

Reviewer #2 was the most critical. We are including his/her review in entirety for context above.

"A major earlier criticism related to the identification of 2 categories of neurons, which seemed arbitrary. My concern had been that the two groups of neurons might actually be extreme fractions drawn from a continuous distribution. The fact that the two groups can be separated convincingly by a cluster analysis resolves this doubt. Yet, the result does not mean that these two groups of neurons are in positions allowing them to entertain the subtractive interaction the authors try to advocate. Where are the LPS and the PPS neurons located – are they intermingled, are they found in non-overlapping regions of posterior parietal cortex, perhaps even in different layers, is there any evidence for the kind of connection between them needed to support the suggested subtractive interaction between LPS and PPS neurons? All we learn is that these neurons were recorded from posterior parietal cortex. This is a large regions consisting of a number of well-defined areas. In other words, when I concluded – after having read the previous version of the manuscript – that the authors had studied one of these areas, area LIP, I was obviously wrong. I think what is missing is experimental data addressing the exact anatomical position and the question of the existence of true physiological interactions between LPS and PPS – e.g. based on multielectrode recordings that would allow the authors to look for functional interactions between simultaneously recorded PPS and LPS. Hence, my concern remains that the seeming representation of the error by the subtraction of the two may be an artifact of the many assumptions made like the pretty arbitrary choices of time windows, the choice of saccade amplitude classes etc. For instance, I do not see any a priori reason why the assumed subtractive comparison should be confined to the two time windows chosen by the authors. If the assumed subtractive interaction between PPS and LPS were more than wishful thinking it would be in any case continuous… Moreover, if the population difference really reflected the saccade error, it should represent the error independent of the amplitude of the primary saccade (at least to some extent…). However, unfortunately, also this is not shown. Hence, I would say that the suggested subtractive interaction is a possibility but as yet far from being grounded on a solid experimental footing.

Unfortunately, also my criticism of the guiding concept and its presentation has not been addressed convincingly. Actually, I feel a bit guilty having drawn the author's attention to work on the cerebellum in processing motor errors and adjusting motor behavior. The reason is that considerations of the cerebellum now take a lot of space in both the Introduction and the Discussion, yet without really contributing to the question why – in the first place – there may be need for a parietal representation of saccadic errors. My original point had been that the Robinson saccade model and any of the many alternatives we have seen over the years work successfully with an internal feedback circuit controlling saccades, circuitry that is purely subcortical. The desired saccade amplitude is represented by the SC – and perhaps LIP etc. Yet, I do not see why the desired saccade amplitude representation in cortex requires error feedback. The authors are mixing up the need to adapt the efference copy in a forward model based on sensory feedback and the question if there are any consequences for the original saccade plan. I am not saying that there cannot be a reason why cortex may want to be informed about the execution of the saccade plan. But all the evidence argues against a role in the online control of saccades. The authors need a clear concept – e.g. related to differences between short term and long-term learning, metalearning etc. Convincing considerations in this direction are completely lacking. I do not think that lengthy – and partially fallacious – considerations of motor error representations in cerebellar cortex can compensate this deficiency.

A few further concrete remarks:

In the third paragraph of the subsection “The perisaccadic activity of PPS neurons reflects the intended eye position”, correlations between perisaccadic activity and postsaccadic position are being presented. Why analyze the horizontal and vertical components independently?

In the fourth paragraph of the subsection “The perisaccadic activity of PPS neurons reflects the intended eye position”, the authors conclude that reward expectation was not correlated with the postsaccadic activity because the reward delivered did not change. Reward expectation also depends on internal processes that vary. Hence, this conclusion is not justified.

In the subsection “The intended eye position was encoded by the relative change, rather than the absolute firing rate, of the perisaccadic activity of PPS neurons”, when comparing the results from the SCS and MGS task, the authors conclude that the”intended eye position was not task specific". This is based on comparing activity of PPS neurons in a narrow time window. However, there is no reason to assume that the monkey´s intention would have been confined to this window. And at earlier times the discharge was clearly very different.

In the first paragraph of the subsection “The activity of LPS neurons reflects extraocular proprioceptive signals”, we read that only 6 out of 22 LPS neurons had a significant correlation with eye position. This is hard to reconcile with the obviously significant effects of eye position discussed later with respect to Figure 5. Any explanation?

In the first paragraph of the subsection “The activity of LPS neurons reflects extraocular proprioceptive signals”: the distribution is clearly skewed. Hence, a t-test is not applicable.

In the second paragraph of the subsection “The activity of LPS neurons reflects extraocular proprioceptive signals”: the latency of LPS responses is on average 70.9 ms relative to the completion of the saccade. If we add a saccade duration of 30-40 msec, this would mean that the assumed proprioceptive signal would arrive more than 100msec after the beginning of the muscle contraction. I am not convinced that the assumption of a proprioceptive signal is justified. This is way too late for a standard proprioceptive signal and suggests something visual.

[Editors' note: further revisions were requested prior to acceptance, as described below.]

Thank you for resubmitting your work entitled "Neuronal representation of the motor errors in macaque posterior parietal cortex" for further consideration at *eLife*. Your revised article has been evaluated by Eve Marder (Senior editor), a Reviewing editor, and two of the original and very experienced reviewers.

The manuscript has been improved but there are substantial remaining issues that need to be addressed before acceptance, as follows:

As you will see, the first referee, who has seen the paper several times before, continues to have major issues that would normally lead to a rejection of the paper. However, at this advanced stage of the refereeing process, we would like to give you a chance to publish your interesting data, provided you can follow this referee's arguments. Specifically, the referee makes very clear and simple suggestions for a final revision (under the title 'Suggestion'). Provided that you are willing and able to follow these suggestions, and do not introduce new points that might give rise to new issues, we would invite you to do this revision, after which we need to make a final decision about acceptance for publication or not. Ordinarily at this point we would only provide a summary of what remains to be done, but we think you might be curious to see the reviewer's reasoning, as well as his/her suggestions.

*Reviewer #2:*

Let me stress that I very much appreciate the efforts of the authors to improve the manuscript taking comments and criticism into account. However, unfortunately also this now third version of the paper is still far from being flawless. Actually, while some problems have been fixed new ones have been added and the conceptual framework offered is still very poor.

Improved:

The authors now provide reasonable arguments that they may have recorded from LIP, although they should be a bit more cautious when drawing this conclusion in the first paragraph of the Results. I would suggest to add the qualifier”*probably* mostly recorded…" They moreover provide evidence that the two types of neurons (PPS and LPS neurons) were found intermingled. I agree that this may make functional interactions between the two more likely. Yet again, I would phrase this possibility more cautiously.

They also have taken the criticism that the correlation between the postsaccadic error and the subtraction signal may be a fortunate artifact of the time windows chosen. They now show that this result is also obtained if the time windows are changed to some extent. This is an improvement.

Remaining or new problems:

Unfortunately, the problem that the range of target eccentricities tested is limited to 10deg and the range of saccade amplitudes explored therefore being very small remains (see below).

In the fourth paragraph of the subsection “The perisaccadic activity of PPS neurons reflects the intended eye position” the authors deal with the possibility that the early post-saccadic activity of PPS neurons may be a consequence of persistent foveal visual stimulation (first by the fixation spot, then by the fovealized cue/target). In order to show that this is not the case, they resort to a comparison with the memory saccade paradigm in which no peripheral saccade target is fovealized. However, rather than considering the saccade-related activity (which is non-visual) they look at the earlier visual response evoked in this paradigm. Why? I do not see that this addresses the aforementioned question. I would have looked at the saccade-related burst.

The conclusion that the activity of PPS neurons is not correlated with reward is based by them on the fact that reward was constant. This is of course not justified. In order to figure out if reward levels have an impact they need to be varied. They may be allowed to conclude that discharge changes cannot not be due to changes of reward level (as it was constant). But this is not what is said.

In the first paragraph of the subsection “The activity of LPS neurons reflects extraocular proprioceptive signals“, we learn that only 5 out of 22 LPS neurons showed significant correlations between their discharge and saccade amplitudes. This is not a compelling argument for an interest in saccade amplitudes. Only if they had varied target eccentricity (which they did not do) they might have been able to clarify if LPS neurons indeed encode amplitude as claimed. In any case, the next step in the argument is again confusing: in the population they find negative as well as positive correlations with a bias for the latter. This is the basis of the conclusion that the population encodes eye position. The implication of this conclusion is that negative correlations would not be compatible with position encoding, which is not correct. Neurons can encode information in multiple ways, linearly or non-linearly.

Conceptual framework propagated in the Abstract, the Introduction and the Discussion.

Much of what we read in these sections deals with the roles of efference copy signals and the need for proprioceptive feedback. The authors provide evidence for a signal – based on their LPS neurons – that may reflect eye position. Yet, the data does not allow one to decide whether this eye position signal is based on efference copy or proprioceptive feedback. However, already the Abstract talks about the latter. This is simply not correct. I agree that a proprioceptive signal may be possible or even likely in view of its demonstration elsewhere in parietal cortex. Yet, others have provided clear evidence for efference copy signals in posterior parietal cortex (e.g. the Andersen lab for reaching). From my point of view the authors should not make claims that are not justified and discuss the pros and cons honestly in the Discussion. In the Abstract and Introduction though they should be more neutral and use phrases such as eye position related etc. Actually dwelling on the nature of this signal in the Discussion would be much more valuable than the endless pages on motor errors and the role of the cerebellum in the Discussion and already earlier in the Introduction, sentences that do not contribute anything to an understanding of their findings and rather have the flavor of a poor – and occasionally simply wrong – review of a literature that they may not have fully understood. For instance, now we read in the Introduction that error encoding in parietal cortex may be needed in order to deal with the”different state of the extraocular muscles". Well, this is exactly one of the well-established functions of a cerebellar forward model that is able to adjust the cerebellum-dependent efference copy in models of saccade control. And the adjustment is based on the climbing fibre system providing information on the error drawn from the SC as shown convincingly for instance by work coming from the Fuchs lab in Seattle. No need for parietal cortex.

Suggestion:

Why not simply write a very short Introduction in which the authors clearly say that the online control of saccades as well as saccadic learning can in principle be fully explained by well-established computational models that reflect the properties of the brainstem-cerebellum machinery for saccades. No need to go into any details here. Then the authors might say that they studied parietal cortex and unexpectedly found an encoding of eye position and the saccade error. In the Discussion – after a short summary of the major finding and a discussion of the question of the nature of their eye position signal (efference copy vs. proprioception) they could then offer as an explanation of this unexpected finding that parietal cortex may be needed to encode the metrics of the secondary saccade, i.e. the correction saccade. After all, they show that the probability of corrective saccade is predicted by their parietal error signal… The brainstem models I alluded to do not explain how the secondary saccade is planned which is why a cortical contribution might indeed help. However, the authors should carefully check if the timing would be appropriate to encode a secondary saccade whose latency is very short. A second (not exclusive) idea may be that parietal error information might be used to speed up saccadic learning. We know that an understanding of the task and the resulting saccadic errors may help to update the internal image of the target allowing the subject to generate a more appropriate saccade next time. The consequence is that saccadic behavior may change from one trial to the next, rather than requiring many hundred trials if this knowledge is not available.

As said this is a suggestion made in an attempt to help the authors. I very much hope that they will be willing to follow this advice to avoid confusing readers with irrelevant and in many cases false information and ideas.

*Reviewer #3:*

The authors have adequately responded to the reviewers’ comments and have satisfied me. I have no further comments.

---

## [Author Response]

[Editors’ note: the author responses to the first round of peer review follow.]

Reviewer #1:

[…] While interesting, the manuscript has major problems with the presentation, analysis and interpretations. The result leaves a reviewer confused and uncertain exactly what was done, the results and their implications. First is the paradigm. The critical arguments need to be based on the undershooting or overshooting of the target. However, the paradigm was not designed to produce these events, instead the study attempts to obtain that information based on the natural variation in saccade end point position.

Here, we studied the neuronal representation of the comparison between intended and real eye position signals for saccadic error computation. As reviewer 1 mentioned, the study was carried out under natural saccadic eye movements, rather than adaptive saccades (undershooting or overshooting to target). We think that there are at least two advantages comparing natural saccades to adaptive saccades. (1) In natural saccades, the intended eye position is matched with the location of saccadic target because the target is a static goal for an impending saccade. In contrast, the location of saccadic target is displaced after the initiation of a saccade in the adaptive saccades. Therefore, it is not clear whether the intended saccade directs to the pre-saccadic target or the post-saccadic target. (2) In the real life, we make much more nature saccades than adaptive saccades. Thus, studying the neuronal representation of saccadic error computation under nature saccade condition will shed light on understanding the neuronal computation of motor errors.

The analysis assumes that the monkey is always striving for the exact center of the target and, therefore the presaccadic firing does not vary with position but this is a big assumption and needs evidence to support. Without this implicit assumption, the critical analyses in Figure 5 break down. The study would be more convincing if position errors were explicitly controlled. In the present study, saccadic target remained visible in SCS and CCS tasks (Figure 2) while monkeys made saccades. Monkeys were intensively trained for years to make saccade to a small target (0.2°) within a 3° checking window. In fact, our behavioral data showed that the post-saccadic eye position of rewarded trials distributed around the target position similar as a Gaussian distribution (insert histogram in Figure 5). Thus, we believe that, under SCS and CCS task conditions, monkeys intended to make saccades to the location of target. Consistently, we recorded a group of PPC neurons that persistently discharged in both pre- and post-saccadic epochs (PPS neurons), and this perisaccadic activity was not correlated with the real post-saccadic eye position (Figure 3). Therefore, we proposed that the persistent perisaccadic activity of PPS neurons represented the intended eye position to the saccadic target.

This paragraph has been added in the Discussion section paragraph 6.

*The second major problem is the presentation. The study ignores the cerebellum and cerebellum-like literature on sensory prediction error processing. These include the many psychophysical, imaging and patient studies showing that the cerebellum is critical to sensory prediction errors by Shadmehr, Krakauer, and Diedrichsen (see papers such (Diedrichsen et al., 2005; Izawa et al., 2012; Krakauer and Shadmehr, 2006; Shadmehr et al., 2010) (Tseng et al., 2007) as well as single cell studies by Sawtell (Kennedy et al., 2014; Requarth et al., 2014) and Ebner (Popa et al., 2012; Popa et al., 2014).*

Yes. The cerebellum is an important center for generating sensory prediction errors. Such error signals in cerebellum play crucial role in real-time motor control and motor learning. However, it is not clear whether and how the error signals in cerebellum play a role in spatial cognition, such as spatial perception, self-recognition and motor awareness. Previous studies have found that the frontoparietal loop, in particular the posterior parietal cortex (PPC), is crucial for spatial perception, self-recognition and motor awareness. Therefore, we are testing an alternative hypothesis that the PPC is another center for generating the error signals.

Following the comments of the reviewer, we have made substantial changes in the current version of manuscript, including the conceptual model, data analysis and result representation. For instance, we discussed and compared the present study with previous cerebellum literatures in both Introduction and Discussion sections: Introduction paragraph 2, Discussion paragraph 2-4.

Also the presentation of the Results is dominated by description and discussing of supplementary figures. This is somewhat distracting, in some cases the claims are not clear or the supplementary results are not that crucial.

The presentation of the results has been reorganized.

In the Methods, three paradigms were used. However, no data is presented from the color-cue delayed saccade paradigm.

Data of the Color-cue delayed saccade task (CCS) are presented in Figure 5—figure supplement 1A–C.

The third major problem is the analyses and problematic use of statistics. Many claims are based on one-sided t-tests. The vast majority of statisticians agree that one-sided tests should only be used if the alternative is physically impossible. This is not the case here and should not be used.

Sorry for improperly using one-side t-test in the previous version. In the current version, we have changed data analysis by using two-sided t-test. As you can see in the revised text (Results section), even though the p value became little bit larger in two-sided t-test analysis than in one-side t-test analysis, the results did not change qualitatively.

Also, the use of 95% confidence intervals as a measure of population variability is not an appropriate measure nor is SEM a good measure. Instead, standard deviation is a much more appropriate measure of variability and should be used. Also, the single cell analyses in Figure 5 that relate p-values to the correlation coefficients are problematic. First, obviously p-values and the correlation coefficient are highly related and will have somewhat of a bell-shaped profile. This is not informative. Second, performing secondary level statistics on correlation coefficients is not good statistical practice. We agree that the most appropriate measure of the population variability is the standard deviation (STD). However, since neuronal discharge varied vigorously among trials as well as among neurons, practically, 95% confidence intervals (1.96 * SEM) has been frequently used as a measure of population variability or the accuracy of the mean estimation previously (please see the listed publications below). Please note thatnone of the results were calculated based on the activity variance. Actually, we used the 95% confidence intervals to measure the accuracy of the mean estimation.

We agree that the p value and the correlation coefficient are highly related to an individual neuron with monkey’s postsaccadic eye position and the population distribution of a group of neurons will show a bell-shaped profile. The point is if the mean of the bell-shaped distribution of a group of neurons is significantly biased from 0 toward positive value, it means that the discharge of the majority neurons has positive correlation with the postsaccadic eye position.

List of literature using 95% confidence interval to measure the activity variability or the accuracy of the mean estimation:

“A prefrontal–thalamo–hippocampal circuit for goal-directed spatial navigation” (Nature 2015);

“Functional organization of excitatory synaptic strength in primary visual cortex” (Nature 2015);

“Distinct relationships of parietal and prefrontal cortices to evidence accumulation” (Nature 2015);

“Sensory stimulation shifts visual cortex from synchronous to asynchronous states” (Nature 2014);

“Top-Down Versus Bottom-Up Control of Attention in the Prefrontal and Posterior Parietal Cortices” (Science 2008);

“Free choice activates a decision circuit between frontal and parietal cortex” (Nature 2008);

Fourth, there are critical claims in the paper that do not agree with the data presented. In Figure 5, the argument is made that the late post-saccadic activity is linearly related to horizontal eye position based on averaging and binning of the data. It is then claimed in Figure 5 that the same is true for individual cell firing (subsection “The pre- and postsaccadic response neurons behave like a subtraction operator that measure the discrepancy between predicted and actual eye position signals”, second paragraph). However, Figure 5 shows that the p-value for the vast majority of the cells is > 0.05. At best, 2 or 3 cells had a correlation coefficient that reached statistical significance. Furthermore, the correlation coefficients are equally distributed between positive and negative for the included cells in Figure 5, yet there is an overall positive relationship between post-saccadic activity and position in Figure 5. These discrepancies seem very hard to reconcile. While assessing the possibility of the late post-saccadic activity encoding the real post-saccadic eye position, we analyzed the correlation between the post-saccadic activity and the end point of saccade for each later postsaccadic response neurons (LPS) on single trial basis. 22 of 24 LPS neurons were recorded with enough trials (>= 30 trials in the preferred direction) in horizontal saccades. Although only 3 out of 22 individual neurons had a correlation coefficient that reached statistical significance (p<0.05), in population the distribution of the correlation coefficient values was significantly biased to the positive direction (Figure 4, mean r = 0.1028, p=0.0087, paired t-test). Such positive correlation between neuronal activity and end point of saccades indicated that the LPS neurons might encode the real eye position.

Reviewer #2: This manuscript addresses the question of how the error between predictive and actual motor signals are represented in LIP. They propose that one population of neurons 'pre- and post-saccadic neurons' encode the desired target location and another the 'postsaccadic neurons' encode the actual target location, and that the difference between the signals of these two populations encodes the error signal. They also propose that larger error signal correlates with greater probability of corrective saccades. The hypothesis is well described in Figure 1 and the experimental methods are well executed. However, the writing of the manuscript leaves much to be desired. It is often not clear what question is being addressed. Figures are described but what hypothesis is being addressed is left to be inferred by the reader; it would be extremely helpful to first state the hypothesis or prediction at the beginning of each section/paragraph and then describe the data. One important shortcoming is the lack of clarity regarding how the subtracted responses are computed; this is essential for the reader to understand the central tenet of this manuscript. I suspect that all this could be fixed with the help of an experienced writer. As it is, I found it difficult to read and therefore difficult to evaluate. However, if the writing is clarified and the data support the claims made, then it could be a highly impactful paper.

Thanks very much for your kind comments and suggestions. We have changed the manuscript substantially according to your suggestions. Also, the current version of the manuscript was polished by an English speaker.

Comments:

Many English usage errors. e.g. Introduction, first paragraph: ‘…challenged by evidence that indicates…’, subsection “The activity of pre- and postsaccadic response neurons reflects the internal signals to predict the required future eye position”, end of first paragraph: should be “…postsaccadic activity decayed…”? At the start of the second paragraph of the aforementioned subsection: impending?

Thanks; we have corrected these.

In the subsection “The activity of pre- and postsaccadic response neurons reflects the internal signals to predict the required future eye position”, second paragraph: 'in one direction than the other'. Do you mean in the preferred vs. non-preferred direction? Or do you mean in the left direction in the left saccade trials? Figure 3: are all of these 89 from left saccade trials?

We have rewritten this passage to read: “activity in the preferred direction comparing with non-preferred direction”.

*In the subsection “The activity of pre- and postsaccadic response neurons reflects the internal signals to predict the required future eye position”, second paragraph: unclear what the point of this paragraph is. Is it trying to say that pre and postsaccadic activity typically correlated so both could be relevant signal?*

In this paragraph, we want to show that the pre-saccadic activity level and post-saccadic activity level of PPS neuron were highly correlated with each other. Such high correlation between them indicates that the activity in these two intervals might encode similar information.

*In the subsection “The activity of pre- and postsaccadic response neurons reflects the internal signals to predict the required future eye position”, third paragraph: not clear why you are comparing postsaccadic SCS with foveal visual response in MGS. Is this the rationale? "One consideration we excluded is the possibility that the postsaccadic activity is induced by visual stimulation. If so, then one would expect to see response to foveal visual response in the MGS task." However, this rationale is still not very strong. One could have weak visual response that is potentiated more in the SCS task than MGS task.*

In order to exclude the possibility that the postsaccadic activity of PPS neurons is evoked by the foveal visual stimulation, we compared the post-saccadic activity of PPS neurons in SCS with their foveal visual response in MGS. Previous studies have shown that the suddenly onset visual stimulus (salient) appearing in the RF of the LIP neuron will evoke much higher visual response than the situation that a saccade brings a persistent visual stimulus into LIP neuron’s RF. In our study, most of the LIP neurons did not show obvious visual response after foveal visual feedback onset in MGS. This result indicated that the strong post-saccadic response of peri-saccadic response neuron was not evoked by foveal visual stimulation.

In the subsection “The activity of pre- and postsaccadic response neurons reflects the internal signals to predict the required future eye position”, third paragraph: 'post-subtracted activity': what exactly is being subtracted from what? It would be helpful to delineated what portions of response from which neurons fit into the hypothesis outlined in Figure 1.

This is really difficult to parse: 'between the perisaccadic activity of the pre- and post-saccadic neurons and postsaccadic activity of the late postsaccadic response neurons'. Perhaps it would be helpful to shorten 'pre- and post-saccadic neurons' to something like PPS neurons.

We have changed the text accordingly.

Reviewer #3:

The manuscript at question addresses the role of efference copy (=corollary discharge) and proprioceptive information in movement control. The model system considered is visually guided saccades and the hypothesis, the authors present is that parietal area LIP has access to both efference copy information and proprioceptive feedback and moreover, that a particular class of neurons, dubbed presaccadic-postsaccadic (=perisaccadic) response neurons represents the difference between the two. This difference is thought to serve as an error signal, responsible for subsequent corrective movements.

*This argument is based on the distinction of two pools of neurons found in areas LIP of rhesus monkeys. One pool involves neurons which start to fire after a saccade, assumed to represent the acquired eye position after a saccade, reflecting proprioceptive feedback. The second pool consists of neurons with perisaccadic responses thought to represent a prediction of the eye position. The major result is that the late activity of perisaccadic neurons seems to correspond to the difference between their early (presaccadic) response and the discharge of the postsaccadic neurons. Moreover, this difference measure and the late perisaccadic responses thought to represent the difference show an intriguing dependence on the final eye position. The dependence shows a minimum for the ideal position (i.e., for an eye position corresponding to target position) and increases with increasing deviation from this ideal in both directions. This result is intriguing! The dependence found is the one of an error representation suitable to drive corrections and eventually learning. Actually, it seems too specific to be an adventitious consequence of the many assumptions the authors have to make on their way to this particular result, assumptions whose biological significance one may question. Nevertheless, I am not fully convinced that the conclusions drawn are justified and, moreover, that the conceptual framework presented is viable. What are my concerns?* We have changed the text accordingly.

Differentiation of 2 groups of neurons: The authors separate 2 groups basically only based on differences in the amount of presaccadic activity. Yet, this is not sufficient to conclude that neurons falling on either side of the cut off chosen are qualitatively different. The problem associated with this approach can be easily demonstrated by considering the distribution of saccade errors discussed in Figure 5 distribution which is parsed by the authors into a number of error classes. Here they could as well have concluded that saccades found in distinct error classes are qualitatively different. Yet, they assume a continuous distribution. What is needed in order to justify the assumption of separate populations is a rigorous statistical approach (e.g., a cluster analysis), based on many more parameters.

Thanks for your suggestion. We performed the cluster analysis, which showed that PPS and LPS neurons were separated (please see Figure 7).

Author response image 1.We used Gaussian mixture model (neglecting covariance) to classify all neurons (LPS+PPS) into two groups by using expectation maximization.For each single neuron, the mean activity around the saccade (400 ms before, to 400 ms after saccade onset, time bin = 50 ms) was used in the analysis resulting in a 16-dim vector for each neuron. For simple illustration of the 16-dimensional space, we used linear Fisher’s discriminant dimension to project the space onto meaningful dimensions. The two clusters are shown in red and blue, respectively. The ellipses represent the standard deviation of two Gaussians. The clustering result fits our previous neuron grouping (LPS vs. PPS) very well, except that few PPS neurons (red dots, 7 neurons) were classified as in the same cluster as LPS neuron.**DOI:**
http://dx.doi.org/10.7554/eLife.10912.016

Non-visual vs. visual basis of responses: independent of the question if the assumption of two neuronal pools is justified or not, I doubt that the interpretation of presaccadic activity is really as unambiguous as claimed by the authors. For instance, the neuron shown in Figure 3—figure supplement 2 shows a strong build-up of presaccadic activity with a clear peak at the time of the saccade when tested in the SCS paradigm but very little in the MGS paradigm, although probably in both tasks saccades having similar metric were carried out. The major difference between the two is the fact that in the SCS, but not in the MGS paradigm, the saccade target was available all the time. In other words, I would assume that the peripheral visual cue (based on the target) must have contributed significantly to the response in the SCS task. Independent of this specific interpretation: why should a response component, which – according to the authors – reflects the prediction of the saccade-based change in eye position should differentiate between paradigms?

It is well known that the neuronal discharge is task dependent. Previous studies have reported that the pre-saccadic activity of LIP neurons is associated with many cognitive functions, such as spatial attention, saccadic intention, decision-making, etc. It is clear that the pre-saccadic activity in our data does not only represent the desired post-saccadic eye position. Also, the visual target definitely influences the neuron’s activity. Therefore, we claim that the intended eye position is one of the signals that are encoded by the perisaccadic activity of PPS neurons.

Further: why should an efference copy related discharge show the conspicuous build up plus peak associated with the saccade? And why should it begin such a long time before the saccade? My admittedly very subjective intuition is that such profiles reflect an intention and not an efference copy.

We have modified the abstract and Introduction accordingly.. In the previous version we interpreted the persistent pre- and post-saccadic activity as an internal position signal, which might mix intention and efference copy (corollary discharge). Now, we argue that the activity of PPS neurons represents intention for impending saccades, including the intended eye position.

Tuning for eye position (change): If the presaccadic response component of the perisaccadic neurons and the responses of postsaccadic neurons reflected the predicted and experienced saccades, one would expect to see a clear reflection of saccade metrics. The paper does not provide any pertinent information: Do the authors find a tuning for the amplitude of the saccade vector and/or eye position? I think that a clear tuning must be demanded. Moreover, one would like to see that postsaccadic response neurons can be activated by passive eye movements.

Although we did not systematically test the spatial tuning of each neuron to saccade vector and eye position in detail, signal neuron’s activity in preferred and non-preferred direction was compared. Most neurons showed significant difference between these two direction. It indicated that both PPS and LPS neurons had spatial tuning with the post-saccadic eye position. Furthermore, the population activity of LPS neurons was significantly correlated with the real post-saccadic eye position (Figure 5), even though the range of eye position was not very large. We bet that the LPS neurons will be activated by passively moving of the eye.

*Response windows: in order to quantify responses and the subtraction measure, the authors define windows without providing any justification for their specific choices. As they assume that the activity of perisaccadic neurons in the late window reflects the subtraction, nature would have to choose the windows like the authors unless the result were independent of the exact choice of windows. This question relates to the more general one of wiring: is there any biological basis for supporting the assumption that the output of postsaccadic neurons could be subtracted from the discharge of perisaccadic neurons? For instance, could it be that postsaccadic neurons are actually interneurons, potentially revealed by different spike waveforms and discharge statistics?*

We have changed paragraph 15 of the Materials and methods as per your suggestion.

Unfortunately, neuron activity was recorded by an old model of Α-Omega recording system, which did not record the spike wave.

Conceptual concerns: I think that the conceptual framework presented in the Introduction and the Discussion suffers from a lack of clarity and rigor as to the role of parietal cortex in comparing an efference copy and actual eye position. I would guess that probably any oculomotor physiologist believes in the one or the other variant of Robinson saccade model in which the predicted momentary eye position is compared with the desired endpoint position and the difference is driving the saccade. Furthermore, the prediction is continuously updated based on error information, probably by cerebellar signals. This is a brainstem model with the superior colliculus representing the highest level. The neurons representing the prediction (i.e., the efference copy) in this model are the PPRF short-lead burst neurons whose discharge is precisely correlated with saccade kinematics, with a precision of msec. This model works perfectly without involving cortex, which would only disturb because of the unpleasant delays it would contribute. What I am trying to say is that in order to drive saccades, there is no need for the comparison of efference copy and actual eye position at the level of LIP. Such a comparison may take place, yet, the raison d´etre (e.g. perceptual purposes?) the authors have in mind should be presented much more clearly. As yet, I do not see a compelling concept and I do not see a need for control purposes.

We have edited the Introduction (paragraph 2) and Discussion (paragraph 2-4).

[Editors' note: the author responses to the re-review follow.]

The manuscript has been improved but there are some remaining issues that need to be addressed before acceptance, as outlined below:

We apologize for the delay in this decision, which was caused by extensive discussions among the editors and reviewers. Basically, one of the reviewers is strongly supportive of publication, one more negative, and the last midway between. Consequently, there are some remaining concerns that you will need to address in the text before a final decision can be made.

1) If evidence is obtained without a prior working hypothesis – one should at least try to discuss the findings in relation to dominating concepts a posteriori and try to argue why the latter may be insufficient, wrong or whatever. However, to use the standard model to justify the experiments is weak. Please rework the text with this perspective in mind.

Thank you very much for this important suggestion We added the following paragraph to the Introduction to emphasize the insufficiency of cerebellum internal models in motor error detection. Also, we significantly changed the conceptual model in Figure 1.

“However, the cerebellar internal models have disadvantages. Firstly, saccadic commands might be highly variable among saccades with same trajectary, due to the varied task context in which the saccades are made (Shadmehr et al., 2010). […] The involvement of PPC in the adaptaiton of purpose saccades suggests that, independent to cerebellum, there might be external error detector in PPC that compares saccadic plan with sensory feedback signals. Up to date, however, the cortical representation of such external error detection has not been identified.”

2) You assume a specific subtractive mechanism. This requires a specific anatomical relationship and it implies continuous interactions. However, there is no evidence that the strong anatomical assumption is met. You searched in large parts of posterior parietal cortex, ignoring well-defined anatomical boundaries. Secondly, the assumption of error representation is based on the selection of rather arbitrary time windows. Please address in the Discussion whether the assumed error representation could be a serendipitous finding consolidated by later work. However, it might well be an artifact of the many non-substantiated assumptions made.

For the first question, we now present the reconstructed 3-dementional recording map for each monkey based on the recording sites in the Results section. Data of both monkeys show intermixed distribution between two types of neurons (Figure 7), which makes the interaction between them possible. In addition, the memory-guided saccade task (MGS, Figure 2) was used to find the lateral intraparietal area (LIP) based on its biological activity signature: neurons discharged persistently throughout memory (delayed) interval (Andersen and Buneo, 2002; Snyder et al., 1997; Zhang and Barash, 2004). Neuronal data were recorded from the same holes (1 mm in diameter for each hole) of a recording grid in which we recorded persistent response neurons in MGS. The reconstructed recording sites show that all neurons but one were recorded within an area of 3x4 mm^2^ (81 out of 81 neurons) in monkey B and 3x3 mm^2^ (34 out of 35 neurons) in monkey D. Therefore, the PPS and PLS neurons were mostly recorded from LIP.

For the second question, to examine whether the selected window (150-350 ms after saccade end) for error presentation was purely arbitrary, we did a new analysis of the correlation between post-subtraction activity of PPS neurons and saccadic errors by using a 100 ms sliding window with step of 10 ms. The sliding window started at 0 ms and stopped at 450 ms after saccade end. As shown in a new figure (Figure 5—figure supplementary 1), in the population level the post-subtraction activity showed significant positive correlation with saccade error in a relative long interval (100-380 ms after saccade end), in particular in 120-310 ms the correlation was significantly positive (mean R > 0.1) and stable. This paragraph has been added in the Results section.

Also, the following paragraph has been added in the Discussion section, in which we propose future studies to assess the functional significance of the saccadic error signals in PPC:

“In the present study, we showed the correlation of neuronal activity in macaque PPC with the accuracy of primary saccades as well as with the occurrence of corrective (secondary) saccades. […] The positive results indicate that the error signals in PPC is not a simply readout from cerebellum or other oculomotor plant.”

Specific comments for your attention: It is still somewhat of a challenge to get at the robustness of the data. For example, in Figure 5, 18 out of 55 PPS neurons showed a significant position correlation with saccadic error and the population ρ is 0.13. Therefore, less than 2% of the firing variability is error-related. Is this sufficient encoding for the argument? Another example is in Figure 4 in which the correlation between post-saccadic activity of single LPS neurons and eye position is 0.10, or 0.01% of the variability. However, that is an issue best decided by the wider scientific community. We reduced the minimum saccade amplitude of primary saccade from 6 degrees to 4 degrees, in order to include trials with large saccadic error. In Figure 5, 43 out of 56 PPS neurons showed positive correlation between post-subtraction activity and saccadic error, and 19 of these neurons reached statistical significance. Although the averaged mean correlation coefficient (CC) was 0.155, the population post-subtraction activity of PPS neurons was highly correlated with saccadic error (cc = 0.9041, p<0.0001). We do realize that, in our data, the CC value of single PPS neuron’s activity and postsaccadic eye position was relatively low. One reason that caused the lower CC value was the low coding bandwidth of the spike count (discontinued in timing) in single trials compared with relatively continued saccadic errors.

Reviewer #2 was the most critical. We are including his/her review in entirety for context above.

"A major earlier criticism related to the identification of 2 categories of neurons, which seemed arbitrary. My concern had been that the two groups of neurons might actually be extreme fractions drawn from a continuous distribution. The fact that the two groups can be separated convincingly by a cluster analysis resolves this doubt. Yet, the result does not mean that these two groups of neurons are in positions allowing them to entertain the subtractive interaction the authors try to advocate. Where are the LPS and the PPS neurons located – are they intermingled, are they found in non-overlapping regions of posterior parietal cortex, perhaps even in different layers, is there any evidence for the kind of connection between them needed to support the suggested subtractive interaction between LPS and PPS neurons? All we learn is that these neurons were recorded from posterior parietal cortex. This is a large regions consisting of a number of well-defined areas. In other words, when I concluded – after having read the previous version of the manuscript – that the authors had studied one of these areas, area LIP, I was obviously wrong. I think what is missing is experimental data addressing the exact anatomical position and the question of the existence of true physiological interactions between LPS and PPS – e.g. based on multielectrode recordings that would allow the authors to look for functional interactions between simultaneously recorded PPS and LPS.

To assess the anatomical relationship between PPS and LPS neurons, we now present the reconstructed 3-dementional recording map for each monkey based on the recording sites in the Results section. Data of both monkeys show intermixed distribution of two types of neurons (Figure 7), which makes the interaction between them possible. Please see our response to point 2 above.

*Hence, my concern remains that the seeming representation of the error by the subtraction of the two may be an artifact of the many assumptions made like the pretty arbitrary choices of time windows, the choice of saccade amplitude classes etc. For instance, I do not see any a priori reason why the assumed subtractive comparison should be confined to the two time windows chosen by the authors. If the assumed subtractive interaction between PPS and LPS were more than wishful thinking it would be in any case continuous… Moreover, if the population difference really reflected the saccade error, it should represent the error independent of the amplitude of the primary saccade (at least to some extent…). However, unfortunately, also this is not shown. Hence, I would say that the suggested subtractive interaction is a possibility but as yet far from being grounded on a solid experimental footing.* Theoretically, to render the comparison between saccadic plan and sensory feedback possible, the saccadic plan signal should rise up before the initiation of saccade and last until the arrival of the sensory signals after the completion of saccade (External error detector in Figure 1). Based on this assumption, if PPS neurons represent the intended eye position signal, the most important timing is the perisaccadic interval. Therefore, we chose a perisaccadic window (150 ms before to 100 ms after saccade end) to analyze the correlation between the activity of PPS neurons and saccadic errors. On the other hand, the generation of sensory feedback (actual eye position signal) is after the initiation of saccade. To make the comparison between intended and actual eye position signals possible, the comparison should be made before the fully declination of the activity of PPS neurons. Therefore, we selected a time window (25-125ms after saccade end) to analyze the correlation between actual eye position signal and saccadic errors. Although these two windows were chosen artificially, they did represent the critical timing of encoding the intended and actually eye position signals, respectively.

We totally agree that, in theory, there are many possible mathematic models for calculating the comparison of different signals. Here, we proposed a subtraction model for the comparison of intended and actual eye position signals is mainly due to two considerations: simplicity and accuracy. As shown in Figure 5, the subtraction results between intended and actual eye position signals (purple dashed lines) are nicely overlapped with the activity of PPS neurons in a time interval of 150-350 ms after saccade end (black solid lines). Moreover, the subtraction results are linearly correlated with the end position of the primary saccade, i.e., the magnitude of saccadic errors: saccadic end points relative to the goal of saccades (10º) (Figure 5). The magnitude of saccadic errors is highly depended on the end points of the primary saccades, but not dependent on the saccadic amplitudes.

To examine whether the selected window (150-350 ms after saccade end) for error presentation was merely arbitrary, we did a new analysis of the correlation between post-subtraction activity of PPS neurons and saccadic errors by using a 100 ms sliding window with step of 10 ms. The sliding window started at 0 ms and stopped at 450 ms after saccade end. As shown in a new figure (Figure 5—figure supplementary 1), in the population level the post-subtraction activity showed significant positive correlation with saccade error in a relative long interval (100-380 ms after saccade end), in particular in 120-310 ms the correlation was significantly positive (mean R > 0.1) and stable. This paragraph has been added in the Results section. We calculated the correlation between the chosen post-subtracted activity and saccade amplitude for each single neuron in the current manuscript. We found that the correlation of post-subtracted activity with saccade error (mean r = 0.1550) was significantly higher (p=0.0023, paired t-test) than the correlation with the saccade amplitude (mean r = 0.0898)

*Unfortunately, also my criticism of the guiding concept and its presentation has not been addressed convincingly. Actually, I feel a bit guilty having drawn the author's attention to work on the cerebellum in processing motor errors and adjusting motor behavior. The reason is that considerations of the cerebellum now take a lot of space in both the Introduction and the Discussion, yet without really contributing to the question why – in the first place – there may be need for a parietal representation of saccadic errors. My original point had been that the Robinson saccade model and any of the many alternatives we have seen over the years work successfully with an internal feedback circuit controlling saccades, circuitry that is purely subcortical. The desired saccade amplitude is represented by the SC – and perhaps LIP etc. Yet, I do not see why the desired saccade amplitude representation in cortex requires error feedback. The authors are mixing up the need to adapt the efference copy in a forward model based on sensory feedback and the question if there are any consequences for the original saccade plan. I am not saying that there cannot be a reason why cortex may want to be informed about the execution of the saccade plan. But all the evidence argues against a role in the online control of saccades. The authors need a clear concept – e.g. related to differences between short term and long-term learning, metalearning etc. Convincing considerations in this direction are completely lacking. I do not think that lengthy – and partially fallacious – considerations of motor error representations in cerebellar cortex can compensate this deficiency.* Thank you very much for your suggestion. We rewrote the Introduction and added the following paragraph to emphasize the potential role of PPC in saccadic error detection.

“However, the cerebellar internal models have disadvantages. Firstly, saccadic commands might be highly variable among saccades with same trajectary, due to the varied task context in which the saccades are made (Shadmehr et al., 2010). […] The involvement of PPC in the adaptaiton of purpose saccades suggests that, independent to cerebellum, there might be external error detector in PPC that compares saccadic plan with sensory feedback signals. Up to date, however, the cortical representation of such external error detection has not been identified.”

*A few further concrete remarks: In the third paragraph of the subsection “The perisaccadic activity of PPS neurons reflects the intended eye position”, correlations between perisaccadic activity and postsaccadic position are being presented. Why analyze the horizontal and vertical components independently?* There are two components, horizontal and vertical, for the postsaccadic eye position of each oblique saccade. Therefore, when we assessed the correlation between neuronal activity and postsaccadic eye position, we did the correlation coefficient analysis of neuronal activity with horizontal and vertical component, respectively. But, when we assessed the correlation between neuronal activity and saccadic errors, we analyzed the correlation coefficient between neuronal activity and the distance from postsaccadic eye position to target location.

*In the fourth paragraph of the subsection “The perisaccadic activity of PPS neurons reflects the intended eye position”, the authors conclude that reward expectation was not correlated with the postsaccadic activity because the reward delivered did not change. Reward expectation also depends on internal processes that vary. Hence, this conclusion is not justified.* We totally agree that the reward expectation is highly dependent on the internal processes that vary from time to time. However, in our case, because the quality and quantity of reward were identical between trials in which saccades directed to the preferred or the non-preferred direction of the neurons, the different perisaccadic activity of PPS neurons between two saccadic directions (Figure 3) wouldn’t reflect the reward expectation. To make it clearer, we rewrote the following sentence to read:

“Moreover, the perisaccadic activity of PPS neurons was not correlated with reward expectation because the quality and quantity of the reward were identical between trials in which saccades directed to either the preferred or the non-preferred direction of neurons.”

*In the subsection “The intended eye position was encoded by the relative change, rather than the absolute firing rate, of the perisaccadic activity of PPS neurons”, when comparing the results from the SCS and MGS task, the authors conclude that the”intended eye position was not task specific". This is based on comparing activity of PPS neurons in a narrow time window. However, there is no reason to assume that the monkey´s intention would have been confined to this window. And at earlier times the discharge was clearly very different.* It has been known for long time that the discharge of neurons is highly modulated by the difficulty of the tasks (Boudreau et al., 2006; Chen et al., 2008). Indeed, we also found that the same neurons’ absulote firing rate differed remarkably between two tasks—CCS versus MGS (Figure 3—figure supplement 4A–C). Based on findings of previous studies, the presaccadic activity of PPC neurons has been associated with multiple cognitive functions, such as attention, intention, decision making, ect. Here, we ask a specific question whether the perisaccadic activity of PPC neurons represents the intended eye position of an impending saccade. If the answer is positive, we expect to see that the intended eye positon signal remains invariable between different tasks, as long as the saccadic trajactory is same. We found that, despite the alteration in the abslute firing rate of PPS neurons between two tasks, interestingly, relative activity changes during perisaccadic interval remained similar between two tasks (Figure 3—figure supplementary 4D). Therefore, we argue that:

“Thus, the intended eye position was not task specific and it might be more reliably encoded by the relative change of neural activity of PPS neurons, rather than their absolute firing rate.”

In the first paragraph of the subsection “The activity of LPS neurons reflects extraocular proprioceptive signals”, we read that only 6 out of 22 LPS neurons had a significant correlation with eye position. This is hard to reconcile with the obviously significant effects of eye position discussed later with respect to Figure 5. Any explanation? Due to the fact that the spiking activity of signal neurons is noise and the coding bandwidth of the spike count is low, the number of LPS neurons that show significant correlation with eye position is low in our data. Nonetheless, the population distribution of the correlation coefficient values was significantly biased to the positive direction (Figure 4, mean r = 0.1237, p=0.0074, paired t-test). Such positive correlation between neuronal activity and end point of saccades indicated that the population activity of LPS neurons might encode the actual eye position.

*In the first paragraph of the subsection “The activity of LPS neurons reflects extraocular proprioceptive signals”: the distribution is clearly skewed. Hence, a t-test is not applicable.* In the current manuscript, we reduced the minimum saccade amplitude of primary saccade from 6 degree to 4 degree, in order to include trials with large saccade error. Now, the distribution of the CC values between post-subtraction activity and saccadic errors is not significantly different form normal distribution (p=0.2081, Kolmogorov-Smirnov test). In addition, we performed paired Wilcoxon test and the result was similar as the result of t-test.

In the second paragraph of the subsection “The activity of LPS neurons reflects extraocular proprioceptive signals”: the latency of LPS responses is on average 70.9 ms relative to the completion of the saccade. If we add a saccade duration of 30-40 msec, this would mean that the assumed proprioceptive signal would arrive more than 100msec after the beginning of the muscle contraction. I am not convinced that the assumption of a proprioceptive signal is justified. This is way too late for a standard proprioceptive signal and suggests something visual.

There are two reasons for us to believe that the activity of LPS neurons represents extraocular proprioception, rather than vision. First, the LPS neurons did not have visual response in both SCS and MGS tasks (Figure 8). Second, previous studies have reported that the latency of extraocular proprioceptive input in cortex is ~80 ms after the initiation of saccades (Nakamura et al., 1999; Wang et al., 2007).

Author response image 2.The averaged population activity of 27 LPS neurons in the SCS task.Averaged spike density with SEM (shaded area) in the preferred (black) and null direction (grey) are shown separately.**DOI:**
http://dx.doi.org/10.7554/eLife.10912.017

[Editors' note: further revisions were requested prior to acceptance, as described below.]

The manuscript has been improved but there are substantial remaining issues that need to be addressed before acceptance, as follows:

As you will see, the first referee, who has seen the paper several times before, continues to have major issues that would normally lead to a rejection of the paper. However, at this advanced stage of the refereeing process, we would like to give you a chance to publish your interesting data, provided you can follow this referee's arguments. Specifically, the referee makes very clear and simple suggestions for a final revision (under the title 'Suggestion'). Provided that you are willing and able to follow these suggestions, and do not introduce new points that might give rise to new issues, we would invite you to do this revision, after which we need to make a final decision about acceptance for publication or not. Ordinarily at this point we would only provide a summary of what remains to be done, but we think you might be curious to see the reviewer's reasoning, as well as his/her suggestions.

Reviewer #2:

Let me stress that I very much appreciate the efforts of the authors to improve the manuscript taking comments and criticism into account. However, unfortunately also this now third version of the paper is still far from being flawless. Actually, while some problems have been fixed new ones have been added and the conceptual framework offered is still very poor.

Improved:

The authors now provide reasonable arguments that they may have recorded from LIP, although they should be a bit more cautious when drawing this conclusion in the first paragraph of the Results. I would suggest to add the qualifier”probably mostly recorded…"

Following the reviewer’s comment, we now added “probably” before “mostly recorded…”.

*They moreover provide evidence that the two types of neurons (PPS and LPS neurons) were found intermingled. I agree that this may make functional interactions between the two more likely. Yet again, I would phrase this possibility more cautiously.* We changed the sentence as follows: “Such intermingled distribution of PPS and LPS neurons indicated that PPS and LPS were recorded from the same area of the PPC, which may make functional interactions between the two types of neurons more likely.”

*They also have taken the criticism that the correlation between the postsaccadic error and the subtraction signal may be a fortunate artifact of the time windows chosen. They now show that this result is also obtained if the time windows are changed to some extent. This is an improvement.* We thank the reviewer for this positive assessment.

*Remaining or new problems: Unfortunately, the problem that the range of target eccentricities tested is limited to 10deg and the range of saccade amplitudes explored therefore being very small remains (see below).* Indeed, it is very unfortunate that we did not systematically test the neuronal activity with a larger range of the target eccentricities. Nevertheless, we also recorded the behavioral and neuronal data when monkeys made saccades to the visual targets in the oblique direction at 13° eccentricity in the SCS task. The correlation between the neuronal activity and the saccadic error during oblique saccades showed similar results as during horizontal saccades (with 10° eccentricity), as shown in Figure 4 and Figure 4—figure supplement 2. To make this point clearer, we added a new paradigm in Figure 1 to denote the oblique direction, and rewrite the description of SCS task as follows:

“Spatial-cue delayed saccade task (SCS, Figure 1): There are two versions of the SCS task: horizontal (with target at 10° eccentricity) and oblique (with target at 13° eccentricity) During training and data collection, the two versions were presented in separate session…”

*In the fourth paragraph of the subsection “The perisaccadic activity of PPS neurons reflects the intended eye position” the authors deal with the possibility that the early post-saccadic activity of PPS neurons may be a consequence of persistent foveal visual stimulation (first by the fixation spot, then by the fovealized cue/target). In order to show that this is not the case, they resort to a comparison with the memory saccade paradigm in which no peripheral saccade target is fovealized. However, rather than considering the saccade-related activity (which is non-visual) they look at the earlier visual response evoked in this paradigm. Why? I do not see that this addresses the aforementioned question. I would have looked at the saccade-related burst.* I think that the reviewer’s problem arises from our unclear labeling of Figure 2—figure supplement 3 (have been modified in the revised figure). In the figure the memory-guided saccade activity is synchronized on the reappearance of the target (i.e., “feedback on” in Figure 1), now at the fovea. There is no response at this time, although there was a presaccadic burst that ended before the reappearance of the foveal target. This indicates that the perisaccadic activity of the PPS neurons was not a foveal visual response.

*The conclusion that the activity of PPS neurons is not correlated with reward is based by them on the fact that reward was constant. This is of course not justified. In order to figure out if reward levels have an impact they need to be varied. They may be allowed to conclude that discharge changes cannot not be due to changes of reward level (as it was constant). But this is not what is said.* We are grateful to the reviewer for this comment. We now added: “Furthermore, the perisaccadic activity could not be a measure of expected reward. The reward was identical between trials in which saccades directed to either the preferred or the non-preferred direction of neurons. Therefore, the discharge difference between preferred and non-preferred direction (e.g., Figure 2) cannot be due to the level of reward.”

*In the first paragraph of the subsection “The activity of LPS neurons reflects extraocular proprioceptive signals“, we learn that only 5 out of 22 LPS neurons showed significant correlations between their discharge and saccade amplitudes. This is not a compelling argument for an interest in saccade amplitudes. Only if they had varied target eccentricity (which they did not do) they might have been able to clarify if LPS neurons indeed encode amplitude as claimed. In any case, the next step in the argument is again confusing: in the population they find negative as well as positive correlations with a bias for the latter. This is the basis of the conclusion that the population encodes eye position. The implication of this conclusion is that negative correlations would not be compatible with position encoding, which is not correct. Neurons can encode information in multiple ways, linearly or non-linearly.* We are grateful that the reviewer pointed out this issue. We have changed this part of the text as follows:

“…we assessed the possibility whether the activity of LPS neurons could consistently encoding the actual end-position of saccades. If single LPS neurons code (at least partly) linearly for the end-position, one would expect a significant correlation of their activity with the random jitter in the actual saccadic end-position. […] Taken together, although subject to noise on a single neuron basis, our results show that, in contrast to the PPS neurons, LPS neurons' activity was positively correlated with the actual end-position of saccades.”

Conceptual framework propagated in the Abstract, the Introduction and the Discussion.

Much of what we read in these sections deals with the roles of efference copy signals and the need for proprioceptive feedback. The authors provide evidence for a signal – based on their LPS neurons – that may reflect eye position. Yet, the data does not allow one to decide whether this eye position signal is based on efference copy or proprioceptive feedback. However, already the Abstract talks about the latter. This is simply not correct. I agree that a proprioceptive signal may be possible or even likely in view of its demonstration elsewhere in parietal cortex. Yet, others have provided clear evidence for efference copy signals in posterior parietal cortex (e.g. the Andersen lab for reaching). From my point of view the authors should not make claims that are not justified and discuss the pros and cons honestly in the Discussion. In the Abstract and Introduction though they should be more neutral and use phrases such as eye position related etc.

We now changed “proprioception” to “actual end-position of saccade” in the abstract and Introduction.

*Actually dwelling on the nature of this signal in the Discussion would be much more valuable than the endless pages on motor errors and the role of the cerebellum in the Discussion and already earlier in the Introduction, sentences that do not contribute anything to an understanding of their findings and rather have the flavor of a poor – and occasionally simply wrong – review of a literature that they may not have fully understood. For instance, now we read in the Introduction that error encoding in parietal cortex may be needed in order to deal with the”different state of the extraocular muscles". Well, this is exactly one of the well-established functions of a cerebellar forward model that is able to adjust the cerebellum-dependent efference copy in models of saccade control. And the adjustment is based on the climbing fibre system providing information on the error drawn from the SC as shown convincingly for instance by work coming from the Fuchs lab in Seattle. No need for parietal cortex.* We thank the reviewer for his/her helpful comments. We now discuss the nature of the eye position related signals in the Discussion section. We also removed the following from manuscript: “error encoding in parietal cortex may be needed in order to deal with the different state of the extraocular muscles".